# A Closer Look at the Perivascular Unit in the Development of Enlarged Perivascular Spaces in Obesity, Metabolic Syndrome, and Type 2 Diabetes Mellitus

**DOI:** 10.3390/biomedicines12010096

**Published:** 2024-01-02

**Authors:** Melvin R. Hayden

**Affiliations:** Department of Internal Medicine, Endocrinology Diabetes and Metabolism, Diabetes and Cardiovascular Disease Center, University of Missouri School of Medicine, One Hospital Drive, Columbia, MO 65211, USA; mrh29pete@gmail.com; Tel.: +1-573-346-3019

**Keywords:** enlarged perivascular spaces, glymphatic system, metabolic syndrome, obesity, neurovascular unit, perivascular unit, perivascular space, postcapillary venules, precapillary arterioles, type 2 diabetes mellitus

## Abstract

The recently described perivascular unit (PVU) resides immediately adjacent to the true capillary neurovascular unit (NVU) in the postcapillary venule and contains the normal-benign perivascular spaces (PVS) and pathological enlarged perivascular spaces (EPVS). The PVS are important in that they have recently been identified to be the construct and the conduit responsible for the delivery of metabolic waste from the interstitial fluid to the ventricular cerebrospinal fluid for disposal into the systemic circulation, termed the glymphatic system. Importantly, the outermost boundary of the PVS is lined by protoplasmic perivascular astrocyte endfeet (pvACef) that communicate with regional neurons. As compared to the well-recognized and described neurovascular unit (NVU) and NVU coupling, the PVU is less well understood and remains an emerging concept. The primary focus of this narrative review is to compare the similarities and differences between these two units and discuss each of their structural and functional relationships and how they relate not only to brain homeostasis but also how they may relate to the development of multiple clinical neurological disease states and specifically how they may relate to obesity, metabolic syndrome, and type 2 diabetes mellitus. Additionally, the concept and importance of a perisynaptic astrocyte coupling to the neuronal synapses with pre- and postsynaptic neurons will also be considered as a perisynaptic unit to provide for the creation of the information transfer in the brain via synaptic transmission and brain homeostasis. Multiple electron microscopic images and illustrations will be utilized in order to help explain these complex units.

## 1. Introduction

The perivascular unit (PVU) has been recently defined and described by Troili et al. (2020) [1]. They described the PVU as “a key anatomical and functional substrate for the interaction between neuronal, immune, and vascular mechanisms of brain injury, which are shared across different neurological disease” [1]. They defined the PVU in order to emphasize the contributions that are made by both the cellular (structural) and molecular (functional) activities that surround the perforating vessels (pial arteries, arterioles, and precapillary arterioles) and the effluxing vessels (pial postcapillary venules, venules, and veins). This is in addition to their interactions, which determine the function of the normal or benign perivascular spaces (PVS) in health and the pathologic remodeling of dilated or enlarged perivascular spaces (EPVS) that are associated with many neurologic diseases [1]. Much of the focus regarding the PVU, PVS, and dilated EPVS has been on pial arteries, arterioles, and precapillary arterioles. However, this review intends to focus more on the pial postcapillary venules, venules, and veins because the PVU and their PVS serve as the conduit for the recently described glymphatic system (GS) of waste removal that is essential for proper brain homeostasis (Figure 1) [1,2].

PVS are also referred to as Virchow–Robins spaces and are the fluid-filled spaces that ensheathe the pial penetrating vessels, both those entering (arteries, arterioles, precapillary arterioles) and those leaving the brain (postcapillary venules, venules, and veins), as they exit the brain parenchyma back to the subarachnoid space (SAS) allowing the contents of their spaces (interstitial fluid (ISF) and metabolic waste) to eventually enter the cerebrospinal fluid space (CSF) and exit the brain to the systemic circulation (Figure 2) [2].

Throughout this narrative review, the term “true capillary” is utilized in order to distinguish its ultrastructure characteristics from the precapillary arterioles and postcapillary venules that manifest a PVU, which contains the normal PVS and the pathologic dilated EPVS, which associates with pathologic remodeling and many neurologic diseases as in Figure 1. Importantly, the true capillary of the NVU delivers its peripheral blood and cellular contents into the immediately adjacent postcapillary venule perivascular unit (PVU).

The neurovascular unit (NVU) and NVU coupling are well recognized and accepted structural and functional units in the brain, which are responsible for cerebral autoregulation and are essential for the proper maintenance of regional cerebral blood flow (CBF) and brain homeostasis (Figure 2, Figure 3 and Figure 4) [2,3,4,5].

Importantly, the PVU that resides immediately adjacent to the true capillary NVU with its blood–brain barrier (BBB) contains the normal PVS that serves as the conduit for the glymphatic system and the pathological EPVS that are more of an emerging concept that is less well understood (Figure 1, Figure 2, Figure 3, Figure 4, Figure 5 and Figure 6) [1,2,6,7,8,9,10]. 

Troili et al. only recently introduced this newer term (the PVU) in 2020, and since then, there has been increasing interest in this newly defined unit within the postcapillary venule; however, the PVU still remains an emerging topic of study and deliberation (Figure 3, Figure 4, Figure 5 and Figure 6) [1]. Importantly, the PVS have been demonstrated to serve as the construct and the structural conduit for the glymphatic system that is responsible for the clearance of metabolic waste from the interstitial spaces to the SAS and CSF [11,12].

In addition to focusing on pial postcapillary venules, venules, and veins, this narrative review also intends to compare the similarities and differences between the NVU and PVU as well as their structural and functional relationships and how they relate not only to brain homeostasis but also how they relate to the development of enlarged perivascular spaces and clinical neurological disease states as they relate to obesity, metabolic syndrome (MetS), and type 2 diabetes mellitus (T2DM) (Figure 7 and Figure 8) [13,14].

While the NVU and PVU have similar and different functions (Figure 7 and Figure 8), they work collaboratively to maintain the proper functioning of the brain’s vascular and neural systems to provide proper neurovascular coupling to provide homeostatic CBF to provide nutrients and metabolic waste removal [1]. Additionally, the concept and importance of a perisynaptic astrocyte cradle coupling to the neuronal synapses with pre- and postsynaptic neurons will also be considered as a unit to provide brain homeostasis and the creation of information transfer in the brain via synaptic transmission.

Importantly, while the principal focus of this review primarily remains on the pial postcapillary venules and their PVUs with their PVS and EPVS, it is important to note that the PVU and spaces also reside alongside the pia arteries, arterioles, and precapillary arterioles. This arterial system allows for intramural periarterial drainage (IPAD) with removal of metabolic waste to the CSF along the basement membrane(s) (BMs) of the vascular smooth muscle cells and the walls of capillaries and arterioles in a retrograde manner to the SAS [15,16]. Additionally, it is intended to discuss the metabolic waste (MW) clearance from these glymphatic system spaces to the subpial space, SAS, and CSF for delivery into the systemic circulation [1,7,9,10,11]. Also, newer concepts have been emerging regarding the concept that all brain compartments involved in CSF homeostasis are involved with a functional continuous exchange between them rather than just serving as separate fluid compartment receptacles that are primarily based on hydrostatic pressure [6,17]. Accordingly, aquaporin-4 (AQP4) in the pvACef plays a central role in cerebral fluid homeostasis discussed later in greater detail [6,17]. 

In summary, the NVU consists of the following cellular components: neurons, perivascular astrocytes, microglia, pericytes, blood endothelial cell(s) (BECs), and basement membrane(s) (BMs). The cellular components of the NVU form and share intimate, complex interactions and thus are responsible for the formation of a single functional NVU, while the PVU also contains these same cells with the addition of resident perivascular macrophage(s) (rPVMΦs) as depicted in Figure 1C, Figure 6, Figure 7 and Figure 8 [13,14,18]. 

## 2. Obesity, Metabolic Syndrome (MetS), Type 2 Diabetes Mellitus (T2DM), and Global Aging

Obesity, MetS, and T2DM in addition to advanced age are currently global societal problems that are expected to grow over the coming decades [19]. T2DM of this triad and neurodegenerative diseases (including cerebrocardiovascular disease, cerebral small vessel disease (SVD), and thrombotic or hemorrhagic stroke) are also anticipated to develop aging-related EPVS. Currently, the global population is one of the oldest in our history and it is expected to continue to increase over the next 2–3 decades, such that we will observe these four groups to merge and increase in numbers [19].

Obesity with visceral adipose tissue (VAT), MetS, and T2DM predispose to the development of EPVS, impaired synaptic transmission, impaired cognition, and neurodegeneration over time [20,21]; metabolic disorders with MetS are also associated with EPVS [22,23]. 

T2DM is a heterogeneous, multifactorial, polygenic disease that may be characterized by a defect in insulin secretion (the beta cell secretory defect), insulin action (insulin resistance), and chronic hyperglycemia [24]. T2DM is strongly associated with obesity–visceral adipose tissue (VAT), insulin resistance (IR), and MetS, which is known to have numerous, devastating complications including hypertension, vasculopathy (micro-macrovascular disease) with cerebrocardiovascular disease and stroke, peripheral neuropathy, retinopathy and blindness, neuropathy, non-traumatic amputation, and nephropathy. Importantly, T2DM is also associated with dilated EPVS and impaired glymphatic function of interstitial waste (including multiple neurotoxic substances that include misfolded amyloid beta and tau proteins) [25,26,27,28,29]. Additionally, peripheral and brain IR as well as MetS also play an important role in brain remodeling (Figure 9) [22,23,26,30,31].

Notably, T2DM is known to be associated with significant brain remodeling with cognitive impairment and dysfunction (CID), vascular cognitive impairment and dementia (VCID), and the development of EPVS [25,27,28,29,30,31,32,33,34,35,36,37,38]. Interestingly, Fulop et al. examined the brain’s venous system and its role in the development of enlarged perivascular spaces [39]. They reported that while cerebral microbleeds–microhemorrhages are definitely associated with small arterioles and capillaries, there is increasing evidence that rupture of small veins and venules can also result in microbleeds [39]. Cerebral microbleeds–hemorrhages (CMBs) are associated with the rupture of small intracerebral microvessels and associated with impaired neuronal function and have the potential to contribute to cognitive impairment, older-age psychiatric syndromes, and gait disorders [40]. Interestingly, these multifocal CMBs were readily demonstrated in the 20-week-old female, obese, insulin-resistant, diabetic *db/db* preclinical mouse model of T2DM (Figure 10 and Figure 11) [5].

While we were unable to unravel the possible causes for these cerebral microbleeds ultrastructurally, we were able to demonstrate that the regions associated with microbleeds were definitely also associated with the detachment and separation of pvACef, and there was no ultrastructural evidence of them being associated with congenital vascular malformations [5].

It is important to note that during these studies of the female 20-week-old *db*/*db* models, we did not examine the venular systems for EPVS or evidence that venular systems may also be involved with both cerebral microbleeds and microinfarcts, since we were not aware of their importance at that time (2018); if you are not looking for a remodeling structure with TEM, you seldom find one [5].

Notably, obesity, MetS, and T2DM have been found to have increased capillary microvascular rarefaction (loss of capillary microvessels) in multiple regions of the brain [2,5,41,42,43,44]. Recently, Shulyatnikova and Hayden hypothesized that capillary microvascular rarefaction might be responsible for the development of EPVS [2]. Capillary microvessel loss due to rarefaction would leave an empty space within the confines of the PVUs’ PVS that would subsequently fill with ISF, and this could allow for an increase in the percent total fluid volume within the PVS that subsequently results in separation of all surrounding pvACef, leaving an EPVS (Figure 12) [2,8]. 

While this capillary microvascular rarefaction still remains a hypothesis that will need to be further tested, it remains an intriguing potential mechanism for the development of EPVS in microvascular disease in the brain. 

## 3. The True Capillary of the NVU Deliverers Its Peripheral Blood and Cellular Contents into the Immediate Adjacent Postcapillary Venule Perivascular Unit

The concept of the NVU and its definition and importance were officially introduced and described in 2001 at the first Stroke Progress Review Group Meeting of the National Institute of Neurological Disorders and Stroke of the National Institutes of Health (NIH) that incorporated neurons and the adjacent vascular cells with the pvACef cells that serve as the connecting cell between the vascular cells and neurons resulting in coupling [4]. The NVU and the NVU coupling are now well accepted and have received great interest in the field of neurobiology. Its cells are comprised of BECs, Pcs, perivascular astrocyte endfeet (pvACef cells), vascular smooth muscle cells in arterioles and arterial microvessels, and interrogating microglia with the pvACef cells being connected to regional neurons to allow for NVU coupling in order to increase oxygen and nutrients to match neuronal excitement demand [4,45,46,47,48]. Incidentally, the PVU cells are the same as the NVU except for the presence of the phagocytic and antigen presenting resident perivascular macrophage (rPVMΦ) cells in health and disease [7,49]. While the NVU of the true capillary and PVU of the postcapillary venule appear to be an anatomical continuum with many structural similarities and only a few minor differences, their functions seem to be quite different (Figure 7 and Figure 8).

When one reviews Zlokovic’s 2-hit vascular hypothesis for neurodegeneration in Alzheimer’s disease [47], it is unquestionable that there is considerable overlap among risk factors for cerebrovascular disorders including cerebral microvascular disease and dysfunction and late onset Alzheimer’s disease (LOAD), vascular dementia (VaD), neurodegeneration, and impaired cognition [19]. The two-hit vascular hypothesis for Alzheimer’s disease places microvascular disease, and more specifically the NVU BBB, as the first hit of the two-hit hypothesis. Since not only the NVU BBB and the immediately adjacent PVU with its PVS and EPVS are involved, the PVU, PVS, and EPVS could now also be included in this first hit, while the second hit would be the impaired clearance of beta amyloid due to EPVS within the PVU [47].

It has been known for some time that midlife obesity [50], diabetes [51,52], and hypertension [53] are all vascular risk factors that are known to increase the risk for neurodegeneration including LOAD. It is currently well recognized that most cases of LOAD have mixed vascular pathology and SVD [54,55]. Additionally, brain hypoperfusion–hypoxia [56], silent infarcts [57], the presence of one or more infarctions [58], stroke episodes, and transient ischemic or hypoxic attacks all increase the risk of LOAD. Indeed, there may be a continuum of progression in obesity, metabolic syndrome, T2DM to VaD, LOAD, and mixed dementia in addition to the accumulating knowledge that macro-/microvascular disease risk factors might all converge on a common final remodeling disease pathway, involving brain microvascular dysfunction and/or degeneration, as well as amyloid-β and tau pathology [19]. Notably, there has been a trend to soften the once hard-fixed clinical and histopathologic boundary lines drawn between vascular dementia and LOAD. LOAD may be considered to reside under the umbrella of mixed dementias [19]. 

NVU BBB disruption caused primarily by BECact/dys with activated BECs (Hit-1) allows proinflammatory peripheral cytokines/chemokines (pCC) to enter the PVU and the proinflammatory cells to adhere to the aBEC within the PVU in addition to allowing increased permeability to multiple neurotoxins from the systemic circulation [59]. The neurotoxic molecules are then delivered to the postcapillary venule’s PVU with its normal PVS and pathologic remodeled EPVS. These neurotoxic molecules and cells with the ensuing metabolic debris begin to accumulate more and more and may result in PVS obstruction, which results in the PVS becoming dilated, enlarged, and remodeled, which results in EPVS that can be identified by non-invasive magnetic resonance imaging (MRI) studies of the brain that are indicative of impaired waste removal via an impaired glymphatic system (Figure 13).

These EPVS (1–3 mm by MRI) can now be identified and quantitated via deep learning algorithms that reduce time, effort, and increase specificity in contrast to the earlier manual hand-counting when viewing MRIs [60].

## 4. The PVU with Its Normal PVS and Pathologic EPVS: Crossroads and Multicellular Crosstalk

Troili et al. were the first to define and describe the perivascular unit (PVU) [1], and this concept is taking better hold now that it has also been learned that the glymphatic system utilizes the postcapillary venule PVS as a conduit for MW clearance [1,11,12,61]. The PVU with its normal PVS and pathologic remodeled EPVS is located in the postcapillary venule and is immediately adjacent to the true capillary (Figure 1, Figure 5, Figure 6 and Figure 11). The PVU allows one to visualize the multicellular crosstalk communication possibilities, which includes BECs, Pcs, resident PVMΦs, the outermost delimiting basement membrane (glia limitans) of the pvACef of this unit, and peripheral cellular leukocytes that are able to penetrate the disrupted NVU BBB (Figure 14 and Figure 15). 

The EPVS that result from the multicellular crosstalk and aberrant remodeling are most commonly identified in either the basal ganglia (BG) or the centrum semiovale (CSO) on MRI T2 weighted images. However, EPVS have also been identified in the midbrain (Type III), and they have been identified in the hippocampus and more recently characterized in the subcortical white matter of the anterior superior temporal lobe, cerebellum at the dentate nucleus, and brain stem [62]. With the use of newer more high-intensity 7T MRIs and increased interest in the glymphatic system and EPVS, we will undoubtedly come to learn of even more areas as the venular system is explored and studied more carefully [63].

## 5. Regional Variation in Enlarged Perivascular Space Locations: Basal Ganglion (BG) and Centrum Semiovale (CSO) 

During the remodeling phase in the response to injury wound healing (RTIWH) mechanism [64], multiple remodeling changes occur, including the development of EPVSs, which can be visualized on non-invasive T2-weighted MRI images. These changes are primarily located either in the basal ganglia or the centrum semiovale depending on the clinical disease state of the brain injury and response to injury wound-healing mechanism (Figure 16) [2,64,65,66,67,68].

EPVS are most commonly viewed in the BG and CSO. This observation may represent different pathologic remodeling, in that BG EPVS are more associated with hypertension-related diseases such as SVD, while CSO EPVS are more commonly associated with misfolded protein diseases such as occurs in LOAD, CAA, and CADSIL [69,70,71]. Additionally, with the newer use of algorithm-based identification of EPV we may find new regional variation in other clinical diseases as we perform more refined studies in this field of study. Over time, we have found that this aberrant remodeling and development of EPVS may be also be associated with the venular systems [63] within the brain, since we are learning more and more about the glymphatic system at an exponential rate [72]. 

## 6. Protoplasmic Perivascular Astrocyte Endfeet and Their Aquaporin 4 (AQP4) Water Channels Play a Crucial Role in the Development of Enlarged Perivascular Spaces

### Maintaining the Structural Integrity and Suspension of the Brain

pvACef line the bulk of the brain vasculature at their abluminal surface [9]. Their polarized expression of AQP4 water channels at the plasma membrane of their endfeet are necessary conditions for the functioning glymphatic system pathway of waste removal [11,61]. Actually, Rasmussen, Mestre, and Nedergaard along with Iliff [11,61] were responsible for coining the term glymphatic system that was based on the pvACef and their dependency on the abluminal location of AQP4 at the plasma membrane facing mostly ISF and metabolic waste filled PVS/EPVS at the postcapillary venule without a pial membrane and the CSF filled PVS at the pial-lined precapillary arterioles [11,73]. Loss of AQP4 polarization in the pvACef leads to diminished CSF influx and significantly, a reduced clearance of metabolic waste in the postcapillary venule of the PVU [9,74,75]. 

The role of AQP4 in the maintenance of CNS homeostasis includes proper CSF circulation and flow, potassium buffering, regulation of extracellular space volume, interstitial fluid resorption, neuroinflammation, osmosensing, calcium signaling, cell migration, and importantly, metabolic waste clearance via the PVS/EPVS-glymphatic system [9,76,77]. When AQP4 is dysfunctional or undergoes loss of polarization as occurs in LOAD [68], SVD, VCI, VCID, and VaD [78,79,80], there is dysfunction. Also, when AQP4 is dysfunctional and/or lost as in the clinical diseases neuromyelitis optica [81,82] and neuromyelitis optica spectrum disorders [83,84] as a result of autoantibodies against the AQP4 water channel as well as the genetic knockout rodent models, which are associated with EPVS [11,85], there is dysfunction of the brain. Further, Nielsen et al. were able to demonstrate that nanogold particles staining of AQP4 by transmission electron immunochemistry were localized to the plasma membrane of the pvACef where they tightly adhered to the NVUs’ pvACef basement membrane [86]. I have only had the opportunity to explore the AQP4 by immunohistochemistry in hepatic cirrhosis individuals and included images from the brains of those individuals with encephalopathy and EPVS (Figure 17) [9,64].

Notably, the glymphatic system is most active during sleep when the clearance of exogenous tracers undergoes a doubling of the clearance rate as compared to wakefulness [87]. 

EPVS are most commonly viewed in the BG and CSO. This observation may represent different pathologic remodeling, in that BG EPVS are associated with more hypertension-related diseases such as SVD, while CSO EPVS are more commonly associated with misfolded protein diseases such as occurs in LOAD, CAA, and CADSIL [69,70,71]. Additionally, with the newer use of algorithm-based identification of EPV, we may find new regional variation in other clinical diseases as we perform more refined studies with higher-intensity MRI machines such as 3 and 7T MRIs in this field of study. Over time, we have found that this aberrant remodeling and development of EPVS may be also found to be associated with the venular systems [63] within the brain, since we are learning more and more about the glymphatic system at an exponential rate [72].

## 7. Loss of Polarity of Aquaporin 4 (AQP4) and Dysfunction or Loss of Dystroglycan (DC) Results in Detachment and Separation of pvACef from NVU and the psACef from the Perisynaptic Unit (PSU)

The brain is critically dependent on the homeostatic functions of astrocytes [9]. Astrocytes (AC) are multifunctional and play an essential role in brain development, modeling, and homeostasis [9,88,89]. ACs are among the most abundant cells in the brain and are the master connecting and communicating cells that provide structural and functional support of brain cells at all levels of organization as depicted in Figure 5, in addition to being regarded as the guardians and housekeepers of the brain [9,88]. The large AC cellular presence in the brain and their vast cell–cell communication via gap junctions connexins may be viewed as the brain’s functional syncytium [89,90]. Additionally, their role in controlling volume in the brain is of essential importance in maintaining homeostasis, in which their highly polarized AQP4 water channels provide for the maintenance of the CNS CSF, ISF, PVS, glymphatic space for waste removal, and buoyancy [9,89] in addition to controlling its own size and volume due to its highly AQP4 polarized plasma membranes [9,89]. Homeostatic functions of ACs (via pvACef and perisynaptic ACef) include molecular homeostasis, which includes ion homeostasis of (calcium, potassium, chloride, and potassium), regulation of pH, water transport and homeostasis via AQP 4, and neurotransmitter homeostasis (including glutamate, gamma-aminobutyric acid (GABA), adenosine, and monoamines) for further homeostatic functions [9,88,89]. There are known to be three major types of ACs, which include (1) pvACef and occur primarily in cortical grey matter, (2) fibrous ACs, which occur primarily in cortical white matter, and (3) peripheral astroglial processes (PAPs) important for providing cradling the astrocyte leaflets of the perisynaptic unit [88,89,91].

In this review, the AC focus has been on the pvACef that connect the NVU to neurons responsible for neurovascular coupling and maintain regional CBF, pvACef, and the PAPS or perisynaptic astrocyte endfeet (psACef) that cradle synaptic neurons of the perisynaptic cradle unit (PSU) that control synaptic transmission and information transfer between neurons.

### 7.1. The NVU and PVU pvACef Are Responsible for Neurovascular Coupling and Regional Neuronal Activity-Induced Maintenance of Regional CBF

The pvACef of the NVU and PVUs are known to work in collaborative synergism to maintain the proper functioning of the brain’s vascular and neural systems via coupling to provide homeostatic CBF to provide nutrients as well as metabolic waste removal [1]. Indeed, the NVU and PVU consist of cellular networks that control and maintain BBB integrity and tightly regulate CBF, which is known to match energy supply to neuronal demand (neurovascular coupling) [4,92]. Proper polarization and functioning of both the AQP4 water channels (AQP4) and dystroglycan (DG) at the pvACef are absolutely necessary for the normal functioning of both the NVU and PVU. Once pvACef or perisynaptic astrocytes become dysfunctional or lose their polarization of AQP4 or DG the pvACef will undergo detachment and separation from the NVU BEC and pericyte endfeet (Pcef) BMs and separate, which allows for not only NVU but also PVU dysfunction with increased permeability, and become increasingly vulnerable to the response to brain injury wound-healing mechanisms (Figure 15) [7,9,59,64,89]. In a past study of obese, insulin-resistant, 20-week-old female diabetic *db/db* models, I was able to demonstrate pvACef detachment and separation from the NVU BECs and Pcfps outer BM [7]. This detachment and separation are believed to be caused by the dysfunction or degradation of dystroglycan and *a*4*b*6 integrins and are most likely due to increased oxidative stress (ROS) with matrix metalloproteinase(s) (MMPs)-2, 9 activation with partial or complete degradation (Figure 8, Figure 9, Figure 16, Figure 17, Figure 18 and Figure 19A,B) [7,93].

In summary, the NVU-PVU pvACef are responsible for neurovascular coupling, ensuring that regional CBF is tightly coupled to the activity of neurons in specific brain regions (Figure 17A,B). This dynamic regulation is essential for maintaining optimal brain function and responding to the varying metabolic demands of different brain areas.

### 7.2. Cradling Perisynaptic Astrocyte Endfeet (psACef) Are Responsible for Synaptic Transmission of Information 

psACef play an essential role in cradling neuronal synapses, synaptic transmission, and plasticity (Figure 19C,D and Figure 20) [89,91,94].

Among the numerous roles of the ACs, they are absolutely essential for controlling the volume of CNS, ISF, and PVS within the PVU as well as the AC itself via its highly polarized plasma membrane AQP4 bidirectional water channel. This is especially true at the pvACef in contact with the vasculature as well as including the PVS and the psACef that are in contact with pre- and postsynapses of the PVU [89,91,95,96]. As the psACef form the cradle of the PSU, they also hide the tripartite synapse from nearby regional injurious stimuli, termed “synaptic isolation”, insulation, and shielding (Figure 17C) [91,96]. Further, this pvACef cradling may result in reducing both the “spill-in” of transmitters released during extrasynaptic signaling events and the “spill-out” of transmitters from the synaptic cleft [97]. Importantly, this would contribute to isolating the synapse from the rest of the CNS, i.e., insulating and shielding (Figure 17C).

AQP4 water channel dysfunction, deficiency, and loss of polarization have been demonstrated to show impaired synaptic plasticity and neurotransmission [96,98,99,100]. Notably, there are at least two clinical neurological diseases that are associated with antibodies against AQP4 water channels: neuromyelitis optica [81,82,83] and neuromyelitis optica spectrum disorder [83,84] as previously presented in Section 6. psACef may indeed follow the same detachment and separation (Figure 17C,D) as demonstrated for the pvACef detachment and separation (Figure 8, Figure 9 and Figure 17A,B), since both are dependent on the functioning presence of β dystroglycan and polarized AQP4 water channels that will become aberrant under similar clinical diseases and brain-injurious stimuli [9,94].

Up to this point, I have referred to synapses as the tripartite synapses, but it is important to note that there is strong scientific reason to consider the synapses as tetrapartite synapses, since the extracellular matrix plays such an integral role in synaptic transmission; at the very least, one should refer to these synapses as multipartite synapses [89,101]. 

In summary, the detachment–retraction of pvACef and psACef can both have widespread effects on the NVU and PSU, respectively, and can potentially lead to disruptions in BBB integrity, altered blood flow regulation, compromised metabolic support to neurons, and disturbances in synaptic function. These changes may contribute to neurological disorders and impair overall brain function via neuroinflammation, impaired glymphatic system efflux associated with EPVS, compromised metabolic support to neurons, impaired synaptic function, plasticity, synaptogenesis, and impaired synaptic transmission with impaired transmission of information [9,89,91,94,97].

## 8. The Venular Side of the Perivascular Unit (PVU) and Enlarged Perivascular Spaces (EPVS)

All small vessels including the postcapillary venules, venules, and veins are almost completely covered by astrocytic endfeet that provide for a relative barrier, since there are known to be up to 20 nm clefts/gaps between some pvACef [102]. pvACef ensheathment of cortical postcapillary venules, venules, and veins play an essential role in the transcellular trafficking of metabolic solutes, ions, and water as they diffuse bidirectionally into and out of the interstitial neuronal parenchymal spaces at the level of the PVS within the PVUs and subsequently empty into the SAS and CSF to the systemic circulation (Figure 2) [63,102,103]. Also, the polarized AQP4 water channels located at the plasmalemma of pvACef/psACef are responsible for bidirectional water flow and are essential for proper water and volume homeostasis including the ISF within the PVS and the interstitial spaces of the parenchyma, CNS, CSF, PVS within the PVU, and the astrocyte itself [9,89]. These important functions of the pvACef are necessary and essential for the proper removal of waste by the glymphatic system to the SAS and CSF and systemic circulation previously discussed in Section 7.

The response to injury wound-healing mechanisms due to any brain injury is known to result in the loss of polarization of AQP4 water channels and dysfunction or loss of DG (especially if it is ongoing or chronic as in obesity, MetS, and T2DM due to metainflammation, or age-related disease such as LOAD), which results in dysfunctional regulation of water [9,88,89]. Dysfunction of the AQP4 water channel would allow for the development of both interstitial space edema and enlargement of the PVS with resultant EPVS. In turn, the EPVS would result in increased stalling and stasis within the PVS with the accumulation of neurotoxins, proinflammatory leukocytes, and metabolic debris with the accumulation of neurotoxins that could be delivered to the interstitial space of the neuronal parenchyma. Additionally, this would result in damage to neurons and synapses with the development of impaired cognition and neurodegeneration and the eventual development of various dementias depending on the specific instigating injurious mechanisms. For example, venular pathology has been shown to contribute to vascular dysfunction in LOAD, which results in WMHs and microthrombosis, infarcts, and microbleeds as in Figure 8 and Figure 9 to result in regional ischemia [104,105,106]. Also, single venule blockade in mice models resulted in impaired cerebrovascular structure and function [107,108,109,110]. 

Even though histopathological documentation of venular accumulation of amyloid fragments in both human and animal models has been identified [103,109,110,111,112,113,114,115,116], the role of the venous network and venous dysfunction induced by amyloid accumulation in SVD and CAA will need to be further studied [103].

Notably, Hartmann et al. [107] and Duvernoy et al. [117] have previously described how penetrating venules formed “units” that were surrounded by rings of penetrating arterioles [107]. Exact ratios were not specified in their studies; however, a typical penetrating venule appeared to drain blood supplied by ~4–5 penetrating arterioles (Figure 21) [63,103]. 

Therefore, should the findings of Hartmann et al. [107] and Duvernoy et al. [117] be supported by others, this would help to explain the importance of venular neurovasculome [118] with obstruction–thrombosis to be studied with greater detail since the venular systems have not yet been studied to the degree that the arteriole neurovasculome systems have been studied to date. Notably, pre-clinical studies have confirmed that venular occlusion causes microinfarcts that are remarkably similar to those found in clinical–pathological human studies [119]. Recently, blockage of a single venule in mice increased microinfarcts and vastly impaired cerebrovascular structure and function [107,108]. Importantly, as we continue to study and explore the glymphatic system, we will also advance our understanding of the venular system.

## 9. Conclusions

PVUs with their normal PVS and pathologic remodeled EPVS play essential roles in the development of neuroinflammation, cerebrovascular disease, and neurodegeneration as summarized in Section 9.1, Section 9.2 and Section 9.3.

### 9.1. In Regards to Neuroinflammation

Postcapillary venular EPVS and alterations in the perivascular unit can trigger inflammatory responses in the brain. Disruption of the NVU BBB integrity and increased permeability can allow immune cells and peripheral cytokines/chemokines to infiltrate the brain PVS (Figure 5, Figure 11, Figure 12 and Figure 13), which completes step 1 of Owens’ 2-step process and contributes to neuroinflammation [10]. Once the glia limitans is breeched (step 2 of Owens’ 2-step process) [10], CNS neuroinflammation can exacerbate neuronal damage and contribute to the progression of various neuropathologic disorders. Further, the PVU and its EPVS can release inflammatory mediators that further amplify the neuroinflammatory response via its resident PVMΦs. Thus, the PVU allows a crossroad for extensive cellular crosstalk (Figure 5, Figure 11, Figure 12 and Figure 13) [1].

### 9.2. In regards to Cerebrovascular Disease 

Postcapillary venular EPVS are excessive fluid-filled spaces surrounding blood vessels in the brain and are associated with impaired drainage of interstitial fluid, contributing to changes in cerebral blood flow regulation. Impaired perivascular clearance mechanisms lead to the accumulation of toxic substances (neurotoxins) and contribute to the development of cerebrovascular diseases such as SVD, ischemic and hemorrhagic stroke, and venular thromboses, which are emerging as a risk for the development of microinfarcts, and possibly even accelerated microinfarctions and microhemorrhages or microbleeds that are known to be increased in obesity, MetS, and T2DM (Figure 8 and Figure 9) [5,39]. Rotta et al. found that in human individuals with SVD and LOAD that cerebral microbleeds were common [103]. Further, these microbleeds were associated with the venous vasculature in human individuals with higher resolution 7TMRI [103]. CMBs are a common finding in patients with VSD, CAA, and LOAD [103,120]. Additionally, van Veluw et al. were able to demonstrate that cerebral microbleeds and infarcts commonly co-occurred in CAA utilizing high-resolution 7T MRI that were not identified with either 1.5 or 3T MRI and further identified more microbleeds with 7T [120]. Thus, as we begin to use newer high-resolution MRI, we will begin to advance the knowledge in the field of venular pathology [120]. Also, Morrone et al. proposed that venular amyloid is an important part of both CAA and AD or LOAD pathology [121]. Further, numerous mechanisms related to the pathophysiology of veins, which include EPVS, venous impaired cerebrovascular pulsatility due to the vascular stiffness of aging, and continued studies would allow for further insight into cerebral SVD, CAA, LOAD, and cerebrovascular dysfunction as an associated early remodeling change associated with multiple brain injuries that result in the response to injury wound-healing mechanisms to add further insight [64,120,121] and may lend further insight into the cerebrovascular dysfunction in SVD, CAA, and LOAD [120,121].

### 9.3. In Regards to Neurodegeneration

Postcapillary venular EPVS have been linked to neurodegenerative diseases like LOAD, PD, and MS, and have recently been described to associate with migraine headaches. The associated compromised perivascular drainage may result in the accumulation of beta-amyloid plaques and other neurotoxic substances, which are known to contribute to the progression of neurodegeneration. Accordingly, the PVU, which consists of blood microvessels, pvACef, Pcs with their endfeet, and resident PVMΦs play a crucial role in maintaining the microenvironment for neuronal health. Dysfunction within the PVU with its pathologic EPVS can lead to impaired nutrient supply and waste removal that are known to contribute to neurodegenerative mechanisms. The above three major headings, neuroinflammation, cerebrovascular disease, and neurodegeneration are equivalent to original description of the PVUs by Troili et al. [1] as a result of being the key anatomical and functional substrate for the interaction between immune, vascular, and neuronal mechanisms associated with multiple types of brain injury.

Advancing science in novel areas typically grows on the shoulders of preexisting scientific discoveries and concepts along with advancing technology, such as high-resolution 7T MRI. Since the glymphatic system is totally dependent on the anatomic structure of the fluid-filled perivascular spaces as its conduit, we can expect the growth of knowledge regarding the PVS, EPVS, and the PVU to grow along with this exciting, novel topic of research regarding the GS; and vice versa, as the knowledge expands in the PVS, EPVS, and the PVU field of studies, we may come to understand more about the GS.

A better understanding of the intricate relationships between EPVS, the PVU, and neurological disorders are crucial for developing targeted therapeutic interventions. Strategies aimed at preserving perivascular space function and the promotion of efficient clearance mechanisms hold great promise in mitigating the impact on neuroinflammation, cerebrovascular disease, and neurodegeneration that are associated with each of the clinical diseases mentioned throughout this review.

In summary, it is hoped that this narrative review will help to increase the understanding regarding the importance of not only the microvascular arterial system including its PVS, but also the venular system. The venular system with its PVU and PVS are essential for waste removal including the effective removal of neurotoxins via the glymphatic system that is dependent on the normal functioning of the postcapillary venular PVS as it is known to be the structural conduit for waste removal within the PVU. The normal postcapillary venular PVS are effective in the removal of neurotoxins including soluble oligomeric amyloid beta, tau, other neurotoxic misfolded proteins, and metabolic waste to slow or prevent neurovascular, neuroinflammatory, and neurodegenerative diseases discussed in this review. EPVS identified by MRI are a biomarker of GS stasis and stalled or obstructed waste removal, and are associated with multiple neurodegenerative diseases. Therefore, we must continue to strive to better understand the microvascular venular systems in addition to the arterial systems. Importantly, EPVS can now be identified and quantitated via algorithms that have been created for deep machine learning, which reduce time and effort, and increase specificity in contrast to earlier visual quantifications.

## Figures and Tables

**Figure 1 biomedicines-12-00096-f001:**
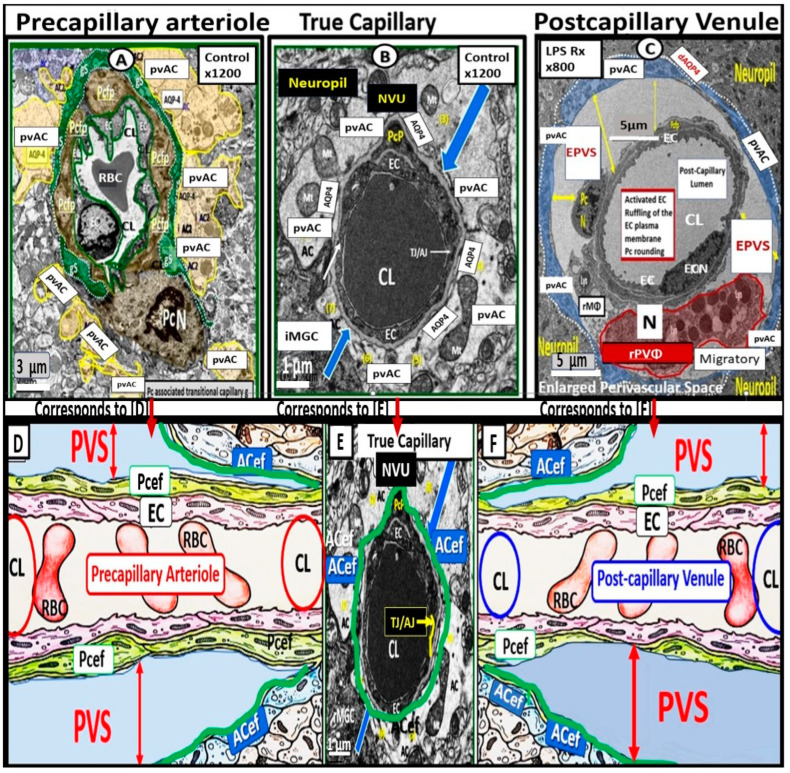
Comparing the ultrastructure transmission electron microscopy (TEM) appearance and illustrations of the true capillary neurovascular unit to the precapillary arteriole and the postcapillary venule with its perivascular spaces (PVS) and enlarged perivascular spaces (EPVS) that exists in the post capillary perivascular unit (PVU). (**A**) A precapillary arteriole, which initially has a larger perivascular space (PVS) at its origin from the subarachnoid space that narrows to a nanometer sized PVS (pseudo-colored green) that ends at the true capillary in the cortical grey and subcortical white matter. (**B**) The primary or true capillary, which has lost the pia mater layer and does not have a PVS. In the true capillary, note how the perivascular astrocyte endfeet (pvACef) tightly abut and are directly adherent to the NVU mural cells, brain endothelial cells (ECs), and pericytes foot processes (PcP–Pcfp) basement membrane(s) (BMs) via the pvACef dystroglycans. The true capillary is the substrate for both the neurovascular unit (NVU) and its blood–brain barrier. Note the interrogating microglia cell (iMGC, white closed arrow). (**C**) A postcapillary venule which is identified via the presence of PVS and in this image depicts a resident perivascular macrophage (rPVMΦ pseudo-colored red). Importantly, the perivascular spaces serve as the construct and structure responsible for carrying the metabolic waste from the interstitial spaces to the cerebrospinal fluid and are known as the glymphatic system pathway that also forms the perivascular unit (PVU). Scale bars = 3 μm; 0.5 μm; 5 μm, respectively. Panels (**A**–**C**) are in cross section and downward red arrows indicate corresponding illustrations in (**D**–**F**). Panels (**D**–**F**) illustrate longitudinal views of the precapillary arterioles, true capillary, and postcapillary venules, respectively, while the cyan green lines represent the glia limitans of the pvACef. Importantly, note the presence of contractile pericytes and their processes in (**C**,**F**) that allow for neurovascular coupling in postcapillary venules. Note that TEM (**A**–**C**) correspond to illustrated (**D**–**F)** respectively and the compressed (**E**) of (**B**) was inserted to illustrate the natural progression of the precapillary arteriole, to the true capillary, and to the postcapillary venule. The modified TEM figures are provided with permission by utilizing the graphic abstract by CC 4.0 [2]. AC = perivascular astrocytes; ACef = astrocyte endfeet—perivascular astrocyte endfeet; AQP4 = aquaporin 4; CL = capillary lumen; dAQP4 = dysfunction aquaporin 4 red lettering; EC = brain endothelial cell; EPVS = enlarged perivascular spaces; gS = glymphatic space—perivascular space; iMGC = interrogating microglial cell; Lys = lysosome; Mt = mitochondria; N = nucleus; NVU = neurovascular unit; Pc = pericyte; PcP = pericyte foot processes; PcN = pericyte nucleus; RBC = red blood cell; rMΦ = reactive macrophage; TJ/AJ = tight and adherens junctions.

**Figure 2 biomedicines-12-00096-f002:**
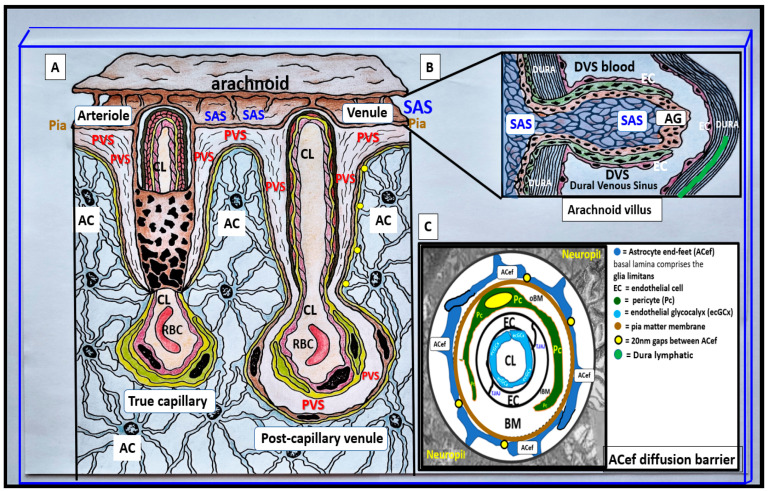
Illustrations of precapillary influx, postcapillary efflux venules, and perivascular spaces (PVS). (**A**) The PVS, which is bounded by the arterial and venous endothelial/pericyte basement membranes and the pia mater/astrocyte endfeet (ACef) (glia limitans) that is responsible for the influx of cerebrospinal fluid (CSF) to the interstitial fluid (ISF) spaces. Likewise, the PVS of postcapillary venules, venules, and veins are responsible for delivery of the ISF admixed with metabolic waste to the SAS and eventually to the systemic circulation via arachnoid granulations. The outermost pia mater abruptly stops at the true capillary and does not exist in the postcapillary venules and veins. Panel (**B**) illustrates the important role of the arachnoid villus and its granulations for exchange of ISF and metabolic waste with the dural venous sinus blood and dural lymphatics (cyan color) and/or the paranasal sinuses (not shown) to reach the systemic circulation bloodstream. (**C**) NVU and the perivascular astrocyte endfeet (pvACef with blue coloring) barrier with a few 20 nm gaps creating a rate-limiting barrier for water and solute exchange. Notably, the pvACef contain the polarized aquaporin 4 (AQP4) water channels, which are known to be important in fluid and solute exchange in addition to the transfer of metabolic waste to the CSF. Note the key in the image. Image provided by CC 4.0 [2].

**Figure 3 biomedicines-12-00096-f003:**
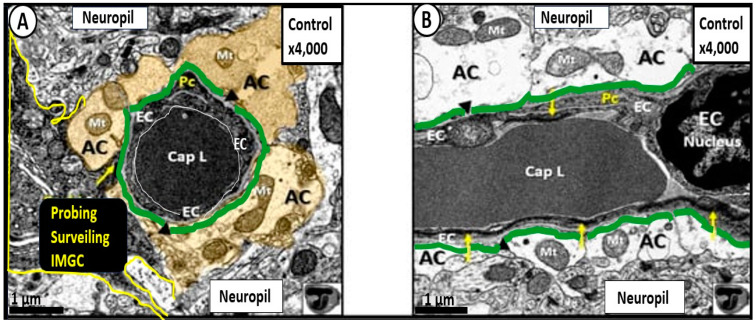
Cross and longitudinal sections of transmission electron microscopic (TEM) images of the true capillary neurovascular unit (NVU). (**A**) TEM cross section of a true capillary NVU. (**B**) True capillary NVU in longitudinal section. The pseudo-colored cyan green line represents and highlights the basement membrane of protoplasmic perivascular endfeet termed the glia limitans perivascularis of the perivascular astrocyte endfeet (**A**,**B**), and the golden pseudo-colored astrocyte endfeet (**A**) represent the critical importance of the perivascular endfeet to the neurovascular unit. Importantly, the pia mater membrane is lost at the level of the true capillary and also the postcapillary venules, venules, and veins of the perivascular unit. Further, note the perivascular astrocyte endfeet (AC) represent the AC clear zone ((**B**), not pseudo-colored as in (**A**)). Modified images provided by CC 4.0 [5]. Magnification x4000; scale bar = 1 μm. AC = perivascular astrocyte endfeet; cap = capillary microvessel; EC = brain endothelial cell; iMGC = interrogating microglia cell; L = lumen; Mt = mitochondria.

**Figure 4 biomedicines-12-00096-f004:**
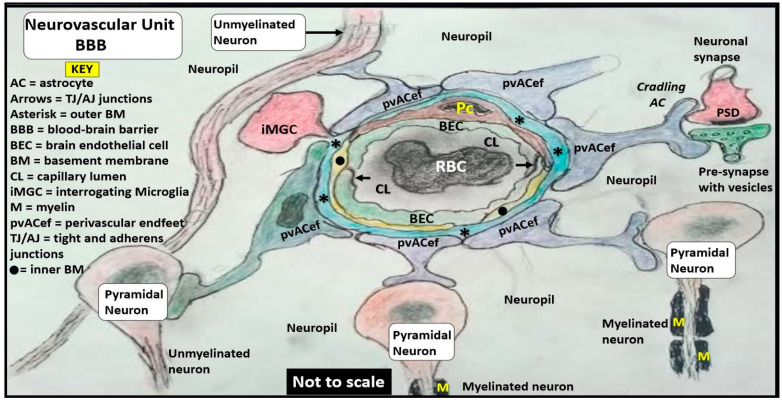
Illustration of the neurovascular unit (NVU) with blood–brain barrier (BBB). Note that not only do the perivascular astrocyte endfeet connect to neurons and dendrites but also to the tripartite synapse (not to scale). Note key for abbreviations within the figure.

**Figure 5 biomedicines-12-00096-f005:**
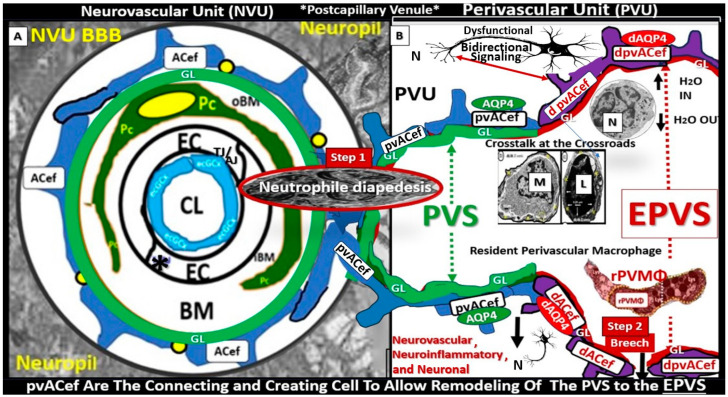
Comparison of the true capillary neurovascular unit (NVU) to the postcapillary venule perivascular unit (PVU). The NVU protoplasmic perivascular astrocyte endfeet (pvACef) (pseudo-colored blue) within the true capillary illustration (**A**) are the connecting and creating cells that allow remodeling of the normal perivascular unit (PVU (**B**)) perivascular spaces (PVS) to transform and remodel into the pathologic enlarged perivascular space (EPVS, which measure 1–3 mm on magnetic resonance imaging). (**A**) Hand-drawn and pseudo-colored control true capillary neurovascular unit (NVU) (representing the transmission electron microscopic (TEM in Figure 1B and Figure 3). Note that when the brain endothelial cells (BECs) become activated and NVU BBB disruption develops, due to BEC activation and dysfunction (BECact/dys) (from multiple causes), there develops an increased permeability of fluids, peripheral cytokines and chemokines, and peripheral immune cells with a neutrophile (N) depicted herein penetrating the tight and adherens junctions (TJ/AJs) paracellular spaces to enter the postcapillary venule along with monocytes (M) and lymphocytes (L) into the postcapillary venule PVS of the PVU (**B**) for step one of the two-step process of neuroinflammation. (**B**) The postcapillary venule that contains the PVU, which includes the normal PVS that has the capability to remodel to the pathological EPVS. Note how the proinflammatory leukocytes enter the PVS along with fluids, solutes, and cytokines/chemokines from an activated, disrupted, and leaky NVU in (**A**). Note how the pvACef (pseudo-colored blue) and its glia limitans (pseudo-colored brown in the control NVU in (**A**) to the cyan color with exaggerated thickness for illustrative purposes in (**B**)) that faces and adheres to the NVU BM extracellular matrix and faces the PVS PVU lumen, since this has detached and separated and allowed the creation of a perivascular space that transforms to an EPVS in (**B**). Also, note how the glia limitans becomes pseudo-colored red, once the EPVS have developed and then become breeched due to activation of matrix metalloproteinases and degradation of the proteins in the glia limitans, which allow neurotoxins and proinflammatory cells to leak into the interstitial spaces of the neuropil and mix with the ISF and result in neuroinflammation (step two) of the two-step process of neuroinflammation [10]. Note that the dysfunctional pvACef AQP4 water channel is associated with the dysfunctional bidirectional signaling between the neuron (N) and the dysfunctional pvACef AQP4 water channel. Image provided by CC 4.0 graphic abstract [9]. AQP4 = aquaporin 4; Asterisk = tight and adherens junction; BBB = blood–brain barrier; BM = both inner (i) and outer (o) basement membrane; dACef and dpvACef = dysfunctional astrocyte endfeet; EC = brain endothelial cell; ecGCx = endothelial glycocalyx; EVPS = enlarged perivascular space; fAQP4 = functional aquaporin 4; GL = glia limitans; H_2_O = water; L = lymphocyte; M = monocyte; N = neutrophile and neuron; Pc = pericyte; PVS = perivascular space; PVU = perivascular unit; rPVMΦ = resident perivascular macrophage; TJ/AJ = tight and adherens junctions.

**Figure 6 biomedicines-12-00096-f006:**
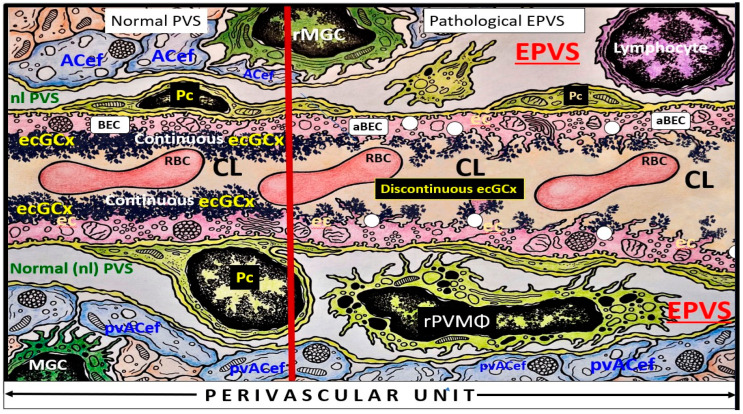
Illustration of the perivascular unit (PVU) with its normal-benign perivascular spaces (PVS) and pathologic enlarged dilated perivascular spaces (EPVS), which lie immediately adjacent to the neurovascular unit true capillary as it transitions from the precapillary arteriole. The vertical red line divides the PVU into the normal PVS on the left-hand side and the pathologic EPVS on the right-hand side of the red line divider. Note the white dots, which represent the attenuation and clumped discontinuous brain endothelial glycocalyx (ecGCx) of the pathologic EPVS in contrast to the continuous ecGCx in the normal PVS. ACef = astrocyte endfeet/pvACef; ecGCx = brain endothelial cell glycocalyx; nl = normal; Pc = pericyte; pvACef = perivascular astrocyte endfeet; RBC = red blood cell; rPVMΦ = resident perivascular macrophage antigen presenting cell; aBEC = activated brain endothelial cell; ACef = astrocyte endfeet/pvACef; ecGCx = brain endothelial cell glycocalyx; nl = normal; Pc = pericyte; pvACef = perivascular astrocyte endfeet; RBC = red blood cell; rMGC = reactive microglial cell; rPVMΦ = resident perivascular macrophage antigen-presenting cell.

**Figure 7 biomedicines-12-00096-f007:**
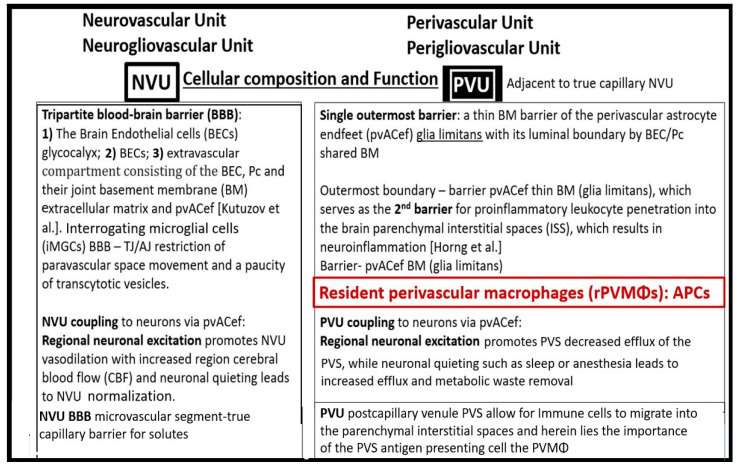
Similarities and differences between the neurovascular unit (NVU) and the perivascular unit (PVU) cellular composition and function. Note the more vulnerable glia limitans—pvACef basement membrane to being breeched and that the PVU has a unique proinflammatory resident PVMΦ (boxed-in red lettering). APCs = antigen-presenting cells; BM = basement membrane; Pc = pericyte; pvACef = protoplasmic perivascular astrocyte endfeet; TJ/AJ = tight and adherens junctions of the blood–brain barrier [13,14].

**Figure 8 biomedicines-12-00096-f008:**
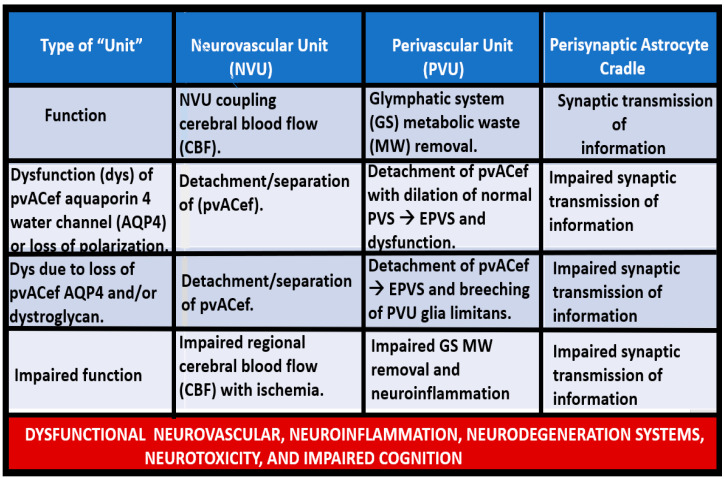
Similarities and differences between function and dysfunction of the neurovascular unit (NVU), perivascular unit (PVU), and the perisynaptic astrocyte endfeet cradle unit (psACef). Note in this table that the psACef and its relation to the tripartite–multipartite synapse with its perisynaptic cradling unit (PSU) may play an important role in the synaptic transmission of information. Further, note that the NVU and PVU function in a collaborative manner to provide homeostatic neurovascular coupling. Kutuzov et al. [13] and Horng et al. [14].

**Figure 9 biomedicines-12-00096-f009:**
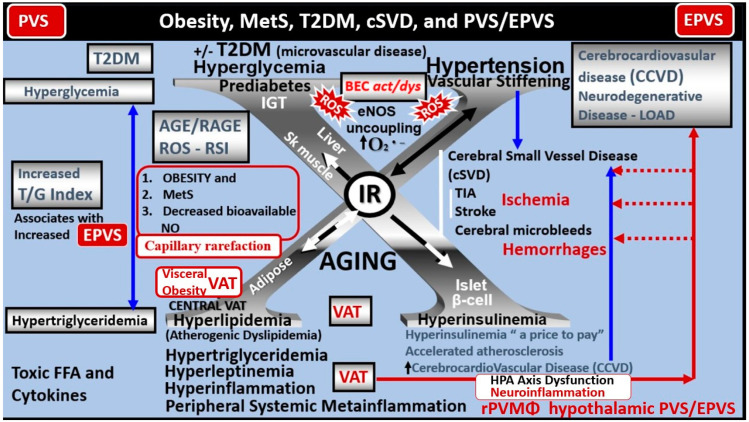
Obesity, metabolic syndrome (MetS), type 2 diabetes mellitus (T2DM), cerebral small vessel disease (SVD), perivascular spaces (PVS), and enlarged perivascular spaces (EPVSs). The visceral adipose tissue (VAT), obesity, and hyperlipidemia (atherogenic dyslipidemia) located in the lower left-hand side of the letter X appears to drive the MetS, peripheral insulin resistance (IR), and brain IR (BIR) that is also located central with the other three arms of the letter X, that includes the associated hyperinsulinemia to compensate for IR (lower right), hypertension and vascular stiffening (upper right), and hyperglycemia (upper left), with impaired glucose tolerance (prediabetes) and with or without manifesting T2DM. Follow the prominent closed red arrows emanating from VAT to cerebrocardiovascular disease (CCVD), SVD, transient ischemic attacks (TIA), stroke, cerebral microbleeds, and hemorrhages. Brain endothelial cell activation and dysfunction (BECact/dys), with its proinflammatory and prooxidative properties, result in endothelial nitric oxide synthesis (eNOS) uncoupling with increased superoxide (O_2_*^•−^*) and decreased nitric oxide (NO) bioavailability in addition to neurovascular unit uncoupling with increased permeability. Importantly, note that obesity, MetS, T2DM, and decreased bioavailable NO interact to result in capillary rarefaction that may allow EPVS to develop, which are biomarkers for cerebral cSVD. While this review does not lend itself to a full discussion of the important role of gut dysbiosis and lipopolysaccharide (LPS) with extracellular vesicle exosomes of LPS producing metainflammation, it was included in this figure. Figure adapted with permission from CC 4.0 [1,8,9,31]. AGE = advanced glycation end-products; RAGE = receptor for AGE; AGE/RAGE = advanced glycation end-products and its receptor interaction; βcell = pancreatic islet insulin-producing beta cell; cSVD = cerebral small vessel disease; FFA = free fatty acids—unsaturated long chain fatty acids; IGT = impaired glucose tolerance; LOAD = late-onset Alzheimer’s disease; ROS = reactive oxygen species; RSI = reactive species interactome; Sk = skeletal: TG Index = triglyceride/glucose index; TIA = transient ischemia attack.

**Figure 10 biomedicines-12-00096-f010:**
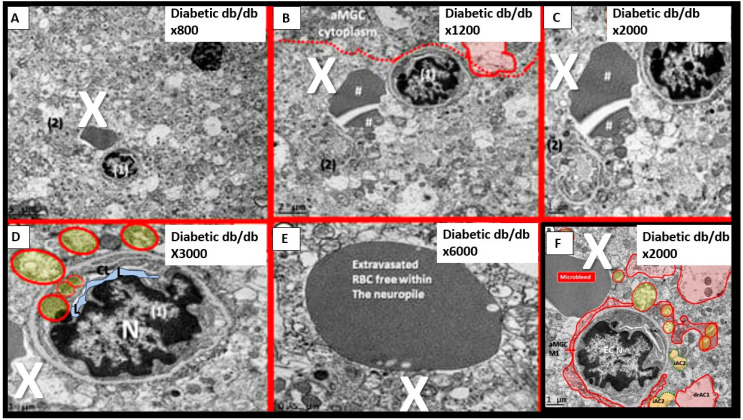
(**A**–**F**) Cerebral microbleeds–hemorrhages in preclinical female obese metabolic syndrome, and type 2 diabetes mellitus genetic models. Each of these six panels depicts a cerebral microbleed identified by a large white X. Note that in (**E**), the homogeneous material may also represent free plasma within the neuropil. Images provided by CC 4.0 [5]. N = nucleus; RBC = red blood cell; X = microbleed; neuropile = neuropil.

**Figure 11 biomedicines-12-00096-f011:**
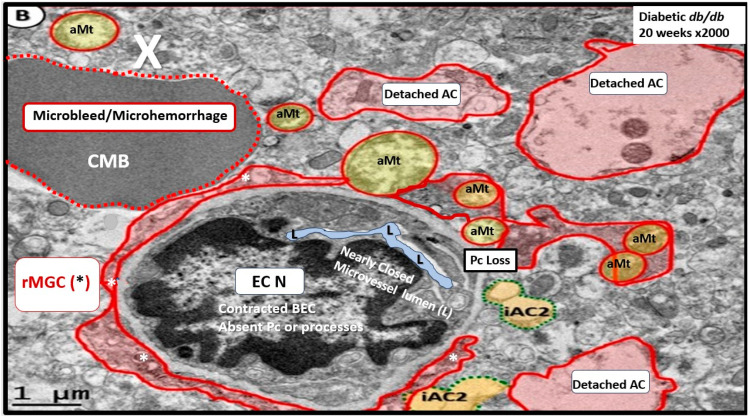
A microbleed (~5 μm) immediately adjacent to a contracted microvessel (~5 μm). Note how the lumen of this microvessel (pseudo-colored light blue) is nearly collapsed and that the brain endothelial cell (BEC) nucleus is contracted with extremely prominent chromatin condensation instead of being heterogeneous, suggesting BEC activation and dysfunction. These similar morphological contracted BEC remodeling changes and nuclear remodeling changes were observed in the aortic endothelium of activated endothelial cells in Western-diet-fed female mice at 20 weeks of age. Also, note that the reactive microglia (pseudo-colored red) encircle this microvessel which contains multiple aberrant mitochondria (aMt), which provide excessive mitochondria-derived reactive oxygen species that provide BEC injury for the response to injury wound-healing mechanisms at the level of this microvessel to result in BEC activation and dysfunction. Importantly, note reactive astrocyte detachment and separation of reactive perivascular astrocytes. These remodeling changes allow for microvessel disruption and microbleeds. Image provided by CC 4.0 [5]. AC = astrocyte; asterisk = reactive microglia cell; CMB = cerebral microbleed; EC N = brain endothelial nucleus; iAC = intact attached astrocyte; rMGC = reactive microglia cell; Pc = pericyte; X = microbleed-microhemorrhage.

**Figure 12 biomedicines-12-00096-f012:**
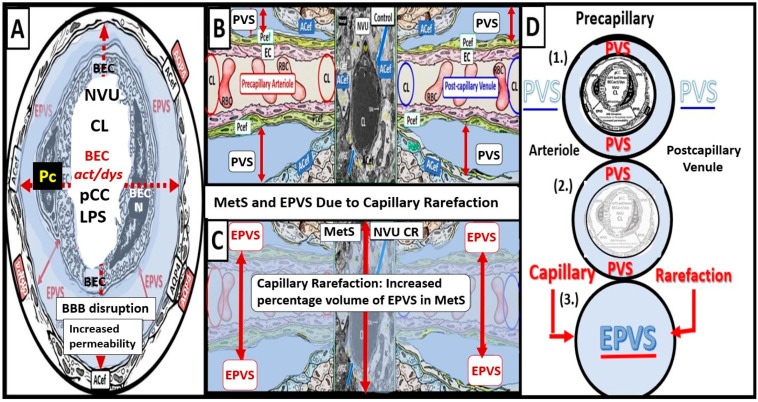
Microvessel rarefaction: Cross and longitudinal sections representative of pre- and postcapillary arterioles and venules with an ensheathing perivascular space (PVS) of the perivascular unit (PVU). (**A**) Cross-section of a capillary microvessel surrounded by PVS (solid double red arrows and light blue color) and its increase in total volume to become an enlarged perivascular space (EPVS) (dashed double red arrows), which represents capillary rarefaction. Note the AQP4 red bars that associate with the perivascular astrocyte endfeet. (**B**) A control longitudinal precapillary arteriole, postcapillary venule, and a neurovascular unit (NVU) capillary that runs through an encompassing PVS (light blue). (**C**) Capillary microvascular rarefaction (CR) in a longitudinal view; note how the volume of the PVS increases its total percentage volume once the capillary has undergone rarefaction as in obesity, metabolic syndrome, and type 2 diabetes mellitus. (**D**) Progression of a normal precapillary arteriole and postcapillary venule PVS to an EPVS once the capillary has undergone rarefaction, allowing for an increase in its total percentage volume of the PVS (1.–3.). (**B**,**C**) provided with permission by CC 4.0 [9]. ACef = perivascular astrocyte endfeet; AQP4 = aquaporin 4 (red bars); BEC = brain endothelial cells; BECact/dys = brain endothelial cell activation and dysfunction; CL =capillary lumen; EC = endothelial cell; lpsEVexos = lipopolysaccharide extracellular vesicle exosomes; NVU = neurovascular unit; Pcef = pericyte endfeet.

**Figure 13 biomedicines-12-00096-f013:**
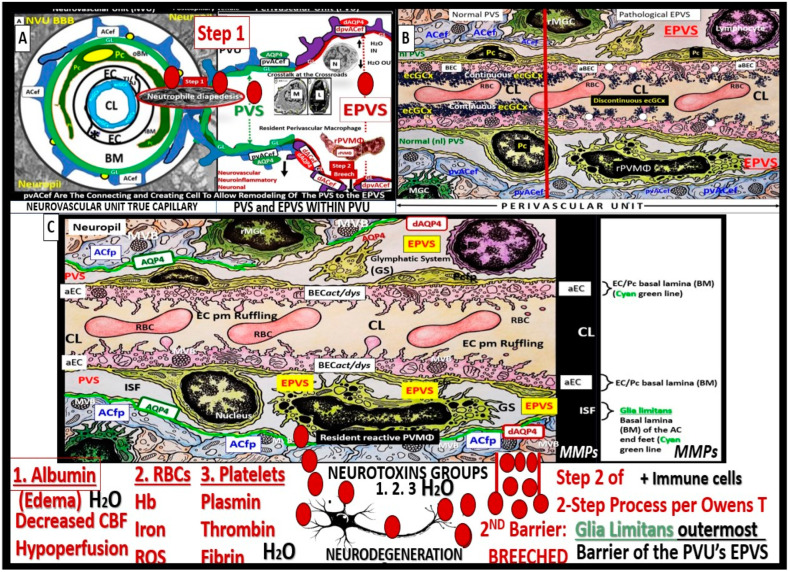
Combining Zlokovic’s 2-hit hypothesis with the neurovascular unit (NVU), perivascular unit (PVU), and the development of enlarged perivascular spaces (EPVS). Panel (**A**) illustrates the NVU true capillary and the PVU with its normal perivascular spaces (PVS) and pathologic EPVS. When the true capillary NVU becomes disrupted, it allows neurotoxins including proinflammatory cytokines/chemokines and proinflammatory cells into the perivascular units’ PVS. Also, note that (**A**) depicts step-1 of Owens 2-step process of neuroinflammation as well as the first hit of Zlokovic’s 2-hit vascular hypothesis [10,47]. Panel (**B**) depicts the PVU with its divisions into the normal PVS and the pathologic EPVS divided by the vertical red line; note the discontinuous endothelial glycocalyx and the presence of the resident perivascular macrophage (rPVMΦ). Panel (**C**) also depicts the PVU; however, its EPVS specifically depicts the breeching of the glia limitans (cyan green) representing step 2 of neuroinflammation [10] as well as the impaired clearance of amyloid beta and accumulation (hit 2) of Zlokovic’s 2-hit hypothesis [47]. Note that the red circles depict various neurotoxin groups that are divided into three groups (1., 2., 3.) that contribute to neuroinflammation, neurodegeneration, and impaired cognition.

**Figure 14 biomedicines-12-00096-f014:**
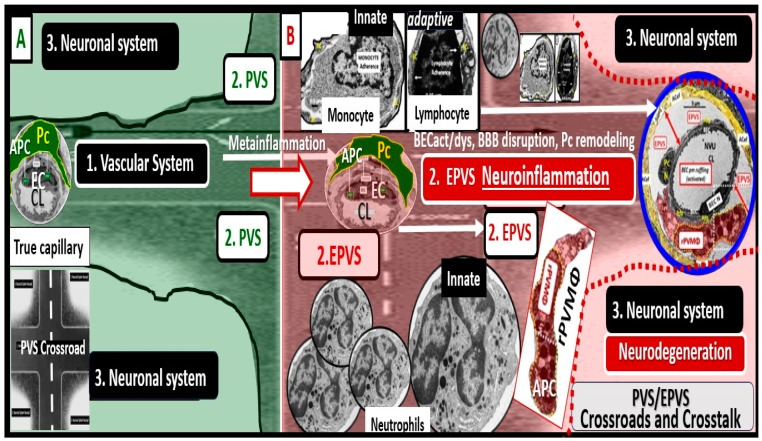
The perivascular unit (PVU) provides a crossroad for multicellular crosstalk communication for vascular, neuroinflammatory, and neuronal systems due to the metainflammation associated with obesity, metabolic syndrome (MetS), and type 2 diabetes mellitus (T2DM). (**A**) Normal-appearing (background pseudo-colored green) perivascular spaces (PVS) indicating their normal function immediately adjacent to the true capillary of the neurovascular unit (NVU). Note the highway intersection icon at the lower left. (**B**) A pathologic enlarged perivascular space (EPVS) (background pseudo-colored red), which suggests pathologic enlargement that resides within the perivascular unit (PVU). Note that this EPVS contains multiple proinflammatory cells (innate immune neutrophils and monocytes, and adaptive immune lymphocytes) that are induced due to the effects of the visceral obesity-associated peripheral inflammation induced at the NVU with BEC activation and dysfunction with increased permeability to allow the proinflammatory cells and neurotoxic cytokines/chemokines to enter the PVS that results in the pathologic remodeling to create EPVS. PVU and EPVS allow for a crossroad or gathering space to form and create the extensive crosstalk communication between the vascular, neuroinflammatory, and neuronal systems to interact to result in neuroinflammation and neurodegenerative changes with resulting impaired cognition that is associated with obesity, MetS, and T2DM. Note that the red-dashed line represents the glia limitans that is breeched to allow step-two of neuroinflammation and subsequent neuronal remodeling. Image made available by CC 4.0 [7]. BBB = blood–brain barrier; BECact/dys = brain endothelial cell activation and dysfunction; CL = capillary lumen; EC = brain endothelial cells; EPVS = enlarged perivascular space; Pc = pericytes; rPVMΦ = resident reactive perivascular macrophage(s).

**Figure 15 biomedicines-12-00096-f015:**
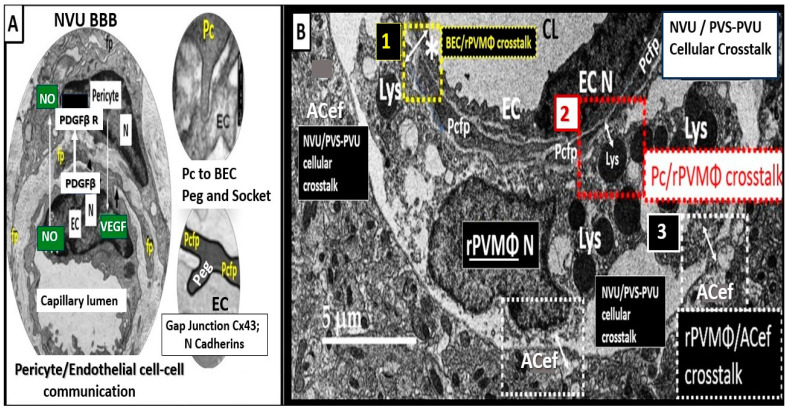
Crosstalk at the crossroads of the perivascular unit (PVU). Multicellular crosstalk between the resident perivascular macrophage (rPVMΦs) and the brain endothelial cells (BECs), pericytes (Pcs), and perivascular astrocyte endfeet (pvACef). (**A**) Normal true capillary of the neurovascular unit (NVU) blood–brain barrier (BBB) interface with peg and socket communicating gap-junctions, connexin 43 (Cx43) and N-cadherin junctions with encircling Pcs, and the cellular signaling utilizing nitric oxide (NO) and platelet-derived growth factor beta (PDGFβ) and vascular endothelial cell growth factor (VEGF). (**B**) Cellular crosstalk between the BECs, Pcs, Pcef, and the pvACef and the rPVMΦs (yellow, red, and white dashed lines, respectively) due to their close proximity within the EPVS. Figure provided by CC 4.0 [7]. ACef = perivascular astrocyte endfeet (pvACef); Asterisk = activated BECs; EC = brain endothelial cell; Lys = lysosomes; N = nucleus; Pcfp = pericyte foot process-endfeet.

**Figure 16 biomedicines-12-00096-f016:**
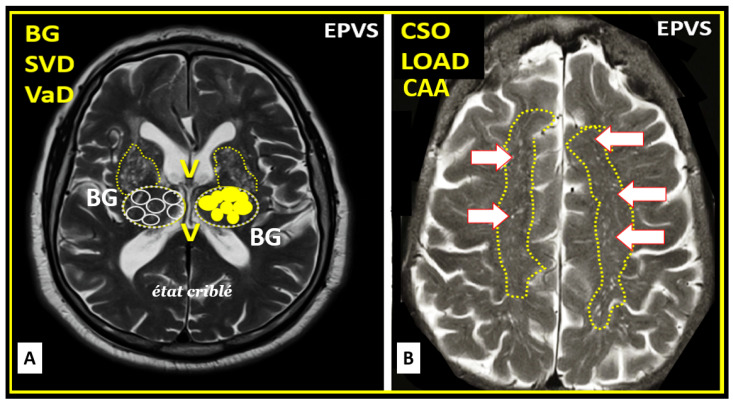
Magnetic resonance imaging (MRI) identification and comparison of basal ganglia (BG) to centrum semiovale (CSO) enlarged perivascular spaces (EPVS). (**A**) Paired EPVSs within the BG that are traced in open circles on the left and solid yellow circles on the right BG. Note the white spaces within the paired dashed lines just above the paired BG structures. MRI image from a 75 year old male post-stroke, recovered with small vessel disease. (**B**) Paired elongated oval structures outlined by yellow dashed lines to enclose multiple white enlarged perivascular spaces. Note the open white arrows outlined in red pointing to prominent EPVSs. MRI image from a 79 year old female with history of transient ischemic attacks. Importantly, note that BG EPVS are strongly associated with cerebral small vessel disease (SVD) in (**A**) and that CSO EPVSs are strongly associated with late-onset Alzheimer’s disease and cerebral amyloid angiopathy (CAA) in (**B**). Incidentally, EPVS are more commonly associated with CSO in atherosclerosis, arteriolosclerosis, obesity, metabolic syndrome, and T2DM. Image reproduced with permission from CC 4.0 [64]. Etat cribble = sieve-like state.

**Figure 17 biomedicines-12-00096-f017:**
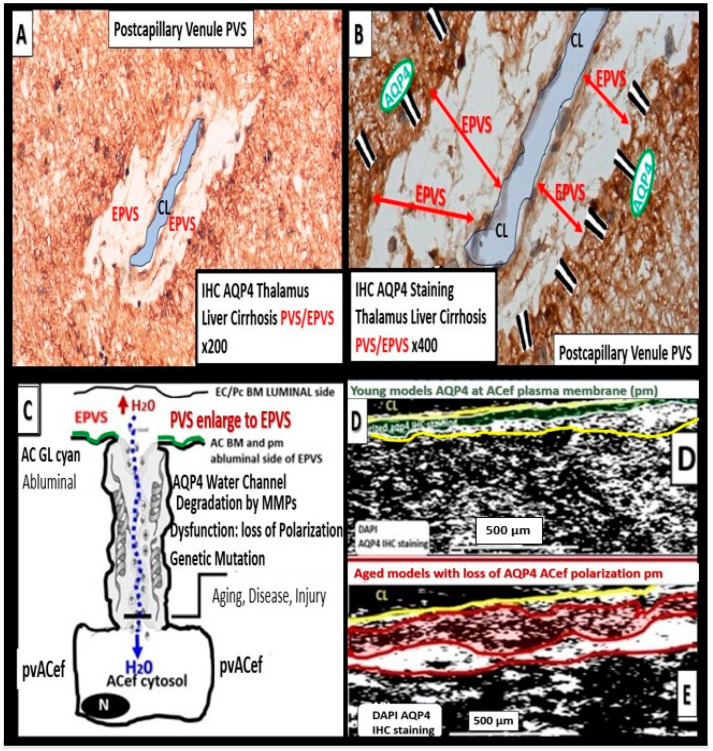
The perivascular astrocyte end feet (pvACef) with its polarized aquaporin 4 (AQP4) water channels delimits the abluminal perivascular unit (PVU) with its perivascular spaces/enlarged perivascular spaces (PVS/EPVS) and perisynaptic astrocyte endfeet (psACef). Panels (**A**,**B**) each demonstrate (via immunohistochemical staining) the presence of AQP4 in the pvACef surrounding a postcapillary venule in an individual with hepatic cirrhosis in the thalamus of the brain. (**C**) Schematic rendering of the AQP4 channel, illustrating water moving into the PVS to contribute the PVS enlargement when AQP4 is dysfunctional and or lost. (**D**) In younger models, AQP4 is tightly polarized to the plasma membrane of the pvACef as compared to (**E**), which depicts a loss of AQP4 polarization in older models. Modified image provided with permission by CC 4.0 [64]. Scale bar = 500 μm. CL = capillary lumen; IHC = immunohistochemistry; N = nucleus.

**Figure 18 biomedicines-12-00096-f018:**
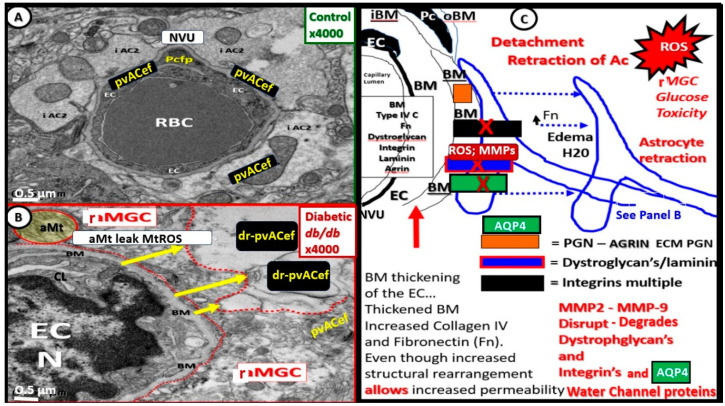
Detachment and retraction of perivascular astrocyte endfeet (pvACef) from the neurovascular unit (NVU) in obese, insulin-resistant, female diabetic *db/db* mice. (**A**) The NVU capillary in control non-diabetic models. Note how the pvACef tightly adhere to the NVU endothelial (EC) and pericyte–pericyte foot processes Pc-Pcfp outer basement membrane (BM). (**B**) Detachment and retraction of reactive pvACef (drpvACef) (yellow arrows) from the NVU. Panel (**C**) illustrates the involved proteins and integrins that are degraded in order for the drpvACef to detach and retract due to increased permeability of the NVU due to NVU disruption. AQP4 = aquaporin 4; AC = astrocyte; BM = basement membrane; EC = brain endothelial cell—endothelium; Fn = fibronectin; MMP2 and MMP 9 = matrix metalloproteinases 2, 9; ROS = reactive oxygen species.

**Figure 19 biomedicines-12-00096-f019:**
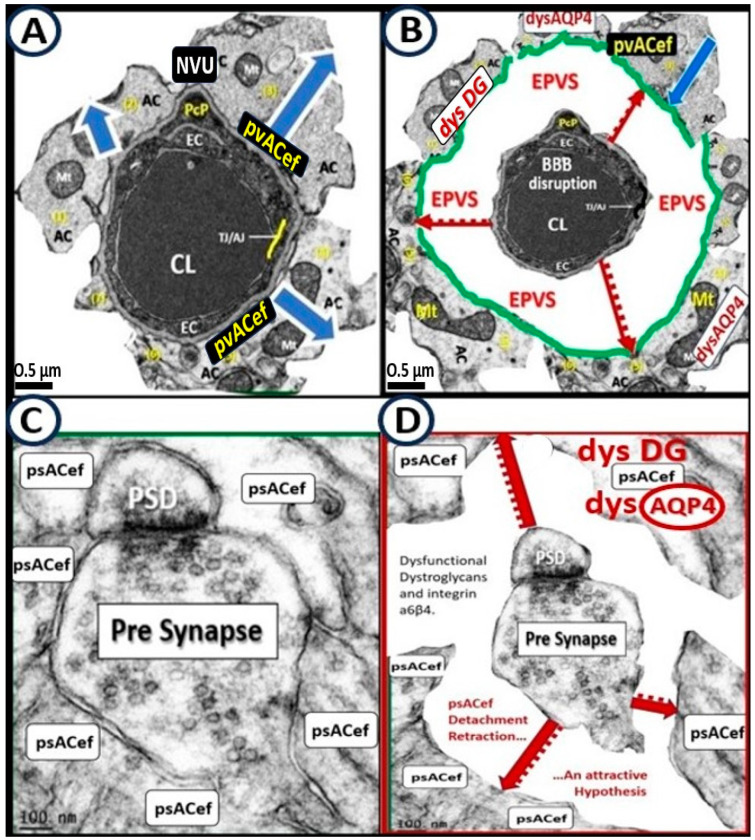
Similarities and comparisons between perivascular astrocyte endfeet (pvACef) and the cradling perisynaptic astrocyte endfeet (psACef) detachment and separation. These similarities implicate damaged or dysfunctional aquaporin 4 (AQP4) either due to activated proteases such as matrix metalloproteinases (MMP-2, 9) or to loss of polarization of AQP4 from the plasma membranes (psACef) resulting in impaired synaptic transmission and impaired cognition or the timing of arrival of incoming information to disturb multiple networks of informational transfer. Panels (**A**,**C**) are from 20-week-old female controls and (**B**,**D**) are from 20-week-old female diabetic *db/db* models with tissues obtained from the frontal cortex, cortical layer III, and depict detachment and separation of pvACef in (**B**) and psACef in (**D**). Note that this detachment and separation creates a perivascular space (PVS) (**B**) and a perisynaptic space (**D**) that may continue to become enlarged with dysfunctional dystroglycan (dysDG) and dysfunctional aquaporin4 (dysAQP4). Note that the cyan green line denoting the glia limitans in (**B**) is not present in (**D**). Images in A and B are reproduced courtesy of CC 4.0 [7]. Scale bars = 0.5 μm in (**A**,**B**) and100 nm (**D**,**E**). BBB = blood–brain barrier; CL = capillary lumen; dys = dysfunctional; DG = dystroglycans; EC = brain endothelial cell; NVU = neurovascular unit; PcP = pericyte process; PSD = post synaptic density; TJ/AJ = tight and adherens junctions.

**Figure 20 biomedicines-12-00096-f020:**
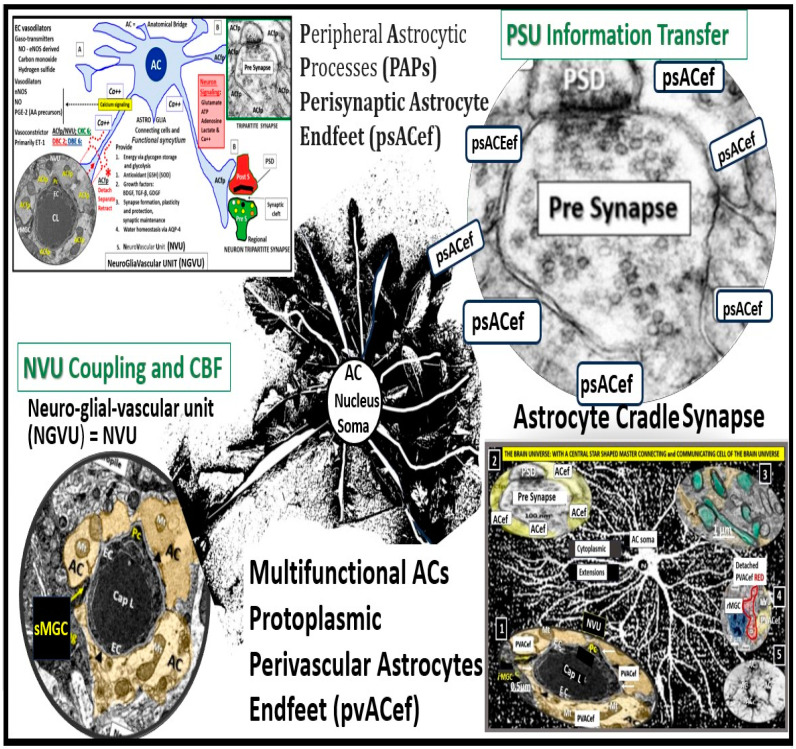
Protoplasmic astrocytes are multifunctional and may be perivascular (perivascular astrocytes endfeet (pvACef)), perisynaptic (perisynaptic astrocyte endfeet (psACef)), or both. This image illustrates a central astrocyte (AC) with its nucleus and soma with multiple protoplasmic extensions. Lower-left image illustrates a transmission electron microscopic (TEM) image of the control true capillary neurovascular unit/neuro-glial-vascular unit (NVU) with a protoplasmic extension connecting with the golden yellow pvACef. Upper-right image illustrates a perisynaptic unit (PSU) with cradling psACef with a protoplasmic AC extension connecting to the psACef that cradle the synapse. Upper-left image illustrates a similar central AC connection to both a NVU and a PSU (image supplied by 4.0 [94]). Lower-right image also illustrates a similar AC connective morphologic image with the central AC connecting to both a NVU and a PSU. Upper-left and lower-left and -right provided by CC 4.0 [94].

**Figure 21 biomedicines-12-00096-f021:**
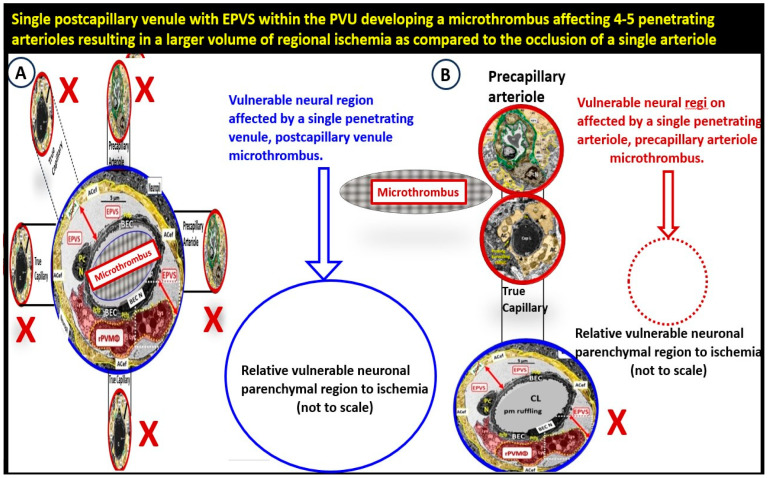
Comparison of a single postcapillary venule microthrombus with enlarged perivascular spaces (EPVS) of the perivascular unit (PVU) compared with a single precapillary arteriole/true capillary microthrombus. (**A**) A single microvessel thrombosis in a postcapillary PVU venule that may affect 4–5 penetrating precapillary arterioles or true capillaries and a much larger vulnerable neuronal region due to decreased cerebral blood flow and ischemia. This also may result in increased microinfarcts, neuronal dysfunction, and even neurodegeneration as compared to the microthrombosis of a single penetrating arteriole as illustrated in (**B**). (**B**) A single precapillary arteriole/true capillary microthrombosis may affect a much smaller vulnerable neuronal parenchymal regional volume as compared to a single venular microthrombosis as in (**A**).

## Data Availability

The data and materials can be provided upon reasonable request.

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
