# Peer review of "A Closer Look at the Perivascular Unit in the Development of Enlarged Perivascular Spaces in Obesity, Metabolic Syndrome, and Type 2 Diabetes Mellitus"

_biomedicines, 2024, doi:10.3390/biomedicines12010096_

Round 1

Reviewer 1 Report

Comments and Suggestions for Authors

This review describes the perivascular unit (PVS) as a relatively new concept and explores changes in its function in patients with different diseases from multiple perspectives, comparing the PVU with another well-known concept, the neurovascular unit (NVU). The relationship between their structure and function is summarized, as well as how they relate to the development of brain homeostasis and various clinical neurological disease states. Most importantly, this paper summarizes their relationship with obesity, metabolic syndrome, and type 2 diabetes, focusing mainly on three aspects: neuroinflammation, cerebrovascular disease, and neurodegeneration. The summary of this paper is comprehensive, covering a wide range, enabling readers to gain a clearer and more comprehensive understanding of the structure, nature, and function of the PVU.

Here are a few aspects the authors could consider further as shown below:

1.    The arrangement depicted in Figure 1 appears somewhat perplexing. In order to enhance clarity, neatness, and aesthetic appeal, it would be beneficial to assign alpha-numeric labels, such as "A", "B" locating above or below each respective small figure. Additionally, it is advisable to ensure that the proportions of the pictures are appropriately adjusted. Notably, Figure E reveals explicit indications of size compression, resulting in a less aesthetically pleasing outcome. Furthermore, Figure 1 contains a combination of electron microscope photos, some displaying a scale while others do not, and Figure 1C, in particular, encompasses two scale formats. This incongruity contributes to an overall disorganized impression. Therefore, it is recommended to consider this feedback when making amendments to the subsequent images.

2.    The article exhibits a clear issue with the picture ratio. To uphold its aesthetics, it is advisable for the author to exercise caution when adjusting the length-to-width ratio of the electron microscope pictures, ensuring they do not appear excessively elongated or excessively short and wide. Prior to the official publish, adjusting the picture ratio is imperative.

3.    The annotation of most figures featured in this article, including figures 1, 2, 5, 6, 7, and 12, appear to be excessively lengthy. To ensure clarity and coherence within the text, it is advisable to provide succinct and concise annotation, while elaborating on the content depicted in the images within the body of the article. This approach will facilitate a more cohesive and logical flow, rather than merely appending pictures with accompanying notes.

4.    When comparing NVU, PVU, and psACef, the contents in the second and third lines of Table 2 are almost the same, but the cells containing contents about NVU and PVU are not mergedwhile the cell of the second and third lines of psACef are merged. Please unify the format.

5.    In the section titled "2. Obesity, Metabolic Syndrome (MetS), Type 2 Diabetes Mellitus (T2DM), and Global Aging," the authors delineate experiments exploring cerebral hemorrhage in mice. Nevertheless, it is important to note that said experiment solely pertained to cerebral hemorrhage resulting from metabolic syndrome and type 2 diabetes, while disregarding cerebral hemorrhage caused by other factors, for instance, congenital vascular malformation, which predominantly manifests in young adults aged between 20 and 40 years, exhibiting a notably high prevalence. Furthermore, the mice utilized in this experiment were comparatively youthful. It would be prudent for the authors to acknowledge the potential relevance of a murine model for cerebral hemorrhage in this disorder.

6.    In order to enhance the overall clarity and coherence of the article, it is recommended to introduce and elucidate the key concepts of "NVU" and "PVU" at the outset. Although the article references the concept of "NVU" multiple times, it is not until the section titled "3. The True Capillary of the NVU Delivers Its Peripheral Blood and Cellular Contents into the Immediate Adjacent 'Postcapillary Venule Perivascular Unit'" that a comprehensive explanation is provided. This delay in clarification may lead readers unfamiliar with these concepts to experience confusion and hinder their understanding of the content. By providing a concise yet comprehensive explanation of "NVU" and "PVU" in the early sections of the article, the overall flow, logical progression, and reader comprehension can be significantly improved.

7.    To enhance the article's organization and facilitate reader comprehension, it is advisable to introduce subheadings under the main headings. These subheadings should provide succinct summaries of the corresponding content, allowing for a clear understanding of the article's key points. For example, within section "6. Protoplasmic Perivascular Astrocyte endfeet and their Aquaporin 4 (AQP4) Water Channels Play a Crucial Role in the Development of Enlarged Perivascular Spaces " subheadings such as "6.1 Maintaining the Structural Integrity and Suspension of the Brain" and "6.2 Maintaining the Normal Shape of the Brain" could be utilized. It is also recommended to refer to the subheadings in section "7. Loss of Polarity of Aquaporin 4 (AQP4) and Dysfunction or loss of Dystroglycan (DC) Results in Detachment and Separation of pvACef from NVU and the psACef from Perisynaptic Unit (PSU) " of the article as a reference for effectively structuring these subheadings. Such an approach will contribute to a more professional and organized presentation of the article's content.

Author Response

Response to Reviewer Number 1 Round 1 for MS ID: biomedicines-2791607

First the author would like to thank reviewer number 1 for the time, effort, and knowledge required to review this submitted manuscript.  Each of the suggested comments and recommended changes were helpful and will make this a better manuscript.  Please note that my changes to the submitted manuscript text (round 1) will be in blue lettering.

Comments and Suggestions for Authors

This review describes the perivascular unit (PVS) as a relatively new concept and explores changes in its function in patients with different diseases from multiple perspectives, comparing the PVU with another well-known concept, the neurovascular unit (NVU). The relationship between their structure and function is summarized, as well as how they relate to the development of brain homeostasis and various clinical neurological disease states. Most importantly, this paper summarizes their relationship with obesity, metabolic syndrome, and type 2 diabetes, focusing mainly on three aspects: neuroinflammation, cerebrovascular disease, and neurodegeneration. The summary of this paper is comprehensive, covering a wide range, enabling readers to gain a clearer and more comprehensive understanding of the structure, nature, and function of the PVU.

Here are a few aspects the authors could consider further as shown below:

  1. The arrangement depicted in Figure 1 appears somewhat perplexing. In order to enhance clarity, neatness, and aesthetic appeal, it would be beneficial to assign alpha-numeric labels, such as "A", "B" locating above or below each respective small figure. Additionally, it is advisable to ensure that the proportions of the pictures are appropriately adjusted. Notably, Figure E reveals explicit indications of size compression, resulting in a less aesthetically pleasing outcome. Furthermore, Figure 1 contains a combination of electron microscope photos, some displaying a scale while others do not, and Figure 1C, in particular, encompasses two scale formats. This incongruity contributes to an overall disorganized impression. Therefore, it is recommended to consider this feedback when making amendments to the subsequent images.

Response to reviewer number 1:

Author wishes to thank reviewer number 1 for this astute observation and recommendation.

1.In the new rebuilt figure, author has now pointed out that illustrative panels DEF correspond to the TEM images ABC respectively and that the compressed image of figure B represented by figure E was inserted to illustrate the natural progression of the precapillary arteriole to the true capillary to the postcapillary venule as follows in blue lettering in legend for figure 1.  Also note that each TEM now has a scale bar.  Author wished to retain the second scale bar in panel C in order for the readers to appreciate the scale of the vascular structures within the perivascular space.  Please note the changes in the rebuilt Figure 1. and the changes in blue lettering in the legend to Figure1.

Figure 1 Legend:  Note that TEM panels A, B, C correspond to illustrated panels D, E, F respectively and the compressed figure E of panel B was inserted to illustrate the natural progression of the precapillary arteriole to the true capillary to the postcapillary venule.  

  1. The article exhibits a clear issue with the picture ratio. To uphold its aesthetics, it is advisable for the author to exercise caution when adjusting the length-to-width ratio of the electron microscope pictures, ensuring they do not appear excessively elongated or excessively short and wide. Prior to the official publish, adjusting the picture ratio is imperative.

Note that author has rebuilt figure 3 such that there is no elongation

  1.  
  2. The annotation of most figures featured in this article, including figures 1, 2, 5, 6, 7, and 12, appear to be excessively lengthy. To ensure clarity and coherence within the text, it is advisable to provide succinct and concise annotation, while elaborating on the content depicted in the images within the body of the article. This approach will facilitate a more cohesive and logical flow, rather than merely appending pictures with accompanying notes.

Author wishes to thank reviewer number 1 for this recommendation; however, author feels that once the reader views the figures that some of these brief discussions are better provided within the text of the figure legend while the figure is fresh on the readers mind and they can easily go back and forth between the figure and the legend, especially in regards to the TEM images.  However, the author will attempt to cut as much as he can in the legends when possible.

  1. When comparing NVU, PVU, and psACef, the contents in the second and third lines of Table 2 are almost the same, but the cells containing contents about NVU and PVU are not merged,while the cell of the second and third lines of psACef are merged. Please unify the format.

Author thanks reviewer number 1 for this important error on my part and therefore this figure was rebuilt and inserted into the revised manuscript in order to unify the format of this table. Please see new table 2

  1. In the section titled "2. Obesity, Metabolic Syndrome (MetS), Type 2 Diabetes Mellitus (T2DM), and Global Aging," the authors delineate experiments exploring cerebral hemorrhage in mice. Nevertheless, it is important to note that said experiment solely pertained to cerebral hemorrhage resulting from metabolic syndrome and type 2 diabetes, while disregarding cerebral hemorrhage caused by other factors, for instance, congenital vascular malformation, which predominantly manifests in young adults aged between 20 and 40 years, exhibiting a notably high prevalence. Furthermore, the mice utilized in this experiment were comparatively youthful. It would be prudent for the authors to acknowledge the potential relevance of a murine model for cerebral hemorrhage in this disorder.

Author thinks this is a great recommendation and note that the author has now inserted a brief paragraph immediately following Figure 9 in blue lettering as follows:

     While we were unable to unravel the possible mechanistic causes for these cerebral microbleeds structurally, we were able to demonstrate that the regions associated with microbleeds were definitely associated with the detachment and separation of pvACef and that there was no ultrastructural evidence of them being associated with congenital vascular malformations.

Regarding incidental information obtained from the company that generated these obese, insulin resistant, and diabetic db/db models, they did not note any prominent findings of congenital vascular malformations during our framwork of this 20-week experiment. mrh     

  1. In order to enhance the overall clarity and coherence of the article, it is recommended to introduce and elucidate the key concepts of "NVU" and "PVU" at the outset. Although the article references the concept of "NVU" multiple times, it is not until the section titled "3. The True Capillary of the NVU Delivers Its Peripheral Blood and Cellular Contents into the Immediate Adjacent 'Postcapillary Venule Perivascular Unit'" that a comprehensive explanation is provided. This delay in clarification may lead readers unfamiliar with these concepts to experience confusion and hinder their understanding of the content. By providing a concise yet comprehensive explanation of "NVU" and "PVU" in the early sections of the article, the overall flow, logical progression, and reader comprehension can be significantly improved.

Author wishes to thank reviewer number 1 for this observation and deficit and bringing it to my attention. I concur with reviewer number 1 and therefore I have now moved the more comprehensive explanation of the NVU and PVU into the first paragraph following Figure 2. As follows in blue lettering:

     Throughout this narrative review, the author has utilized the term “true capillary” in order to distinguish its ultrastructure characteristics from the precapillary arterioles and postcapillary venules that manifests a PVU, which contains the normal PVS and the pathologic dilated EPVS, which associates with pathologic remodeling and many neurologic diseases as in figure 1.  Importantly, the true capillary of the NVU delivers its peripheral blood and cellular contents into the immediately adjacent postcapillary venule perivascular unit (PVU).

  1. To enhance the article's organization and facilitate reader comprehension, it is advisable to introduce subheadings under the main headings. These subheadings should provide succinct summaries of the corresponding content, allowing for a clear understanding of the article's key points. For example, within section "6. Protoplasmic Perivascular Astrocyte endfeet and their Aquaporin 4 (AQP4) Water Channels Play a Crucial Role in the Development of Enlarged Perivascular Spaces " subheadings such as "6.1 Maintaining the Structural Integrity and Suspension of the Brain" and "6.2 Maintaining the Normal Shape of the Brain" could be utilized. It is also recommended to refer to the subheadings in section "7. Loss of Polarity of Aquaporin 4 (AQP4) and Dysfunction or loss of Dystroglycan (DC) Results in Detachment and Separation of pvACef from NVU and the psACef from Perisynaptic Unit (PSU) " of the article as a reference for effectively structuring these subheadings. Such an approach will contribute to a more professional and organized presentation of the article's content.

Response to reviewer number 1:

Author concurs and has made the following insertions in section 6 and please see Section 6 for these changes as follows:

  1. Protoplasmic Perivascular Astrocyte endfeet and their Aquaporin 4 (AQP4) Water Channels Play a Crucial Role in the Development of Enlarged Perivascular Spaces

                              6.1. Maintaining the Structural Integrity and Suspension of the Brain

Protoplasmic perivascular astrocyte endfeet line the bulk of the brain vasculature at their abluminal surface [9]. Their polarized expression of AQP4 water channels at the plasma membrane of their endfeet are necessary conditions for the functioning glymphatic system pathway of waste removal [11, 61].

ALSO

Importantly, maintenance of the brain’s structural integrity and its suspension by being suspended or buoyant state in order to not compress the arterial system and cause decreased CBF or ischemia, which is made possible by the proper regulation of the brains’ water content and distribution [73].

                                 6.2. Maintaining the Normal Shape of the Brain

pvACef play numerous important roles, which are important for controlling volume in the brain, which include the CSF, ISF, PVS, and glymphatic space for waste removal, in addition to controlling its own size and volume due to its highly AQP4 polarized plasma membranes [9, 64].

In section 7. please note the following changes:

In this review the AC focus has been on the pvACef that connect the NVU to neurons responsible for neurovascular coupling and maintain regional CBF, pvACef, and the PAPS or perisynaptic astrocyte endfeet (psACef) that cradle synaptic neurons of the perisynaptic cradle unit (PSU) that control synaptic transmission and information transfer between neurons.

7.1. The NVU and PVU pvACef Are Responsible for Neurovascular Coupling and Regional Neuronal Activity-Induced Maintenance of Regional CBF

The pvACef of the NVU and PVUs are known to work in collaborative synergism to maintain the proper functioning of the brain's vascular and neural systems via coupling to provide homeostatic CBF to provide nutrients as well as metabolic waste removal [1]. 

Also:

In summary, the NVU-PVU pvACef are responsible for neurovascular coupling, ensuring that regional cerebral blood flow is tightly coupled to the activity of neurons in specific brain regions (Fig. 17A, B).  This dynamic regulation is essential for maintaining optimal brain function and responding to the varying metabolic demands of different brain areas.

7.2. Cradling Perisynaptic Astrocyte Endfeet (psACef) Are Responsible for Synaptic Transmission of Information

psACef play an essential role in cradling neuronal synapses, synaptic transmission, and plasti city (Fig. 17C, D), and 18) [89, 91, 94].

As an author, it always is exciting for me to see how a reviewer’s suggested changes and recommendations to a submitted manuscript can make it better and improve the submitted manuscript.  Author is in hopes that these changes will now make the revised manuscript acceptable for publication.

Thank you very much!

Melvin R Hayden

Submitting author

Reviewer 2 Report

Comments and Suggestions for Authors Impressive review: following are my few suggestion to improve the manuscript.

Figure 1, footnotes: why do you use EC for brain endothelial cell, and later in the text you use BEC? Be consistent with the acronyms.

Line 122, “BBB”: explain this acronyms here, that is the first time you use it in the manuscript body (no matter if you have explained it in the footnotes of figure 4).

Line 183, “MetS and T2DM”: explain these acronyms here (not in line 225), that is the first time you use them in the manuscript body.

Line 220, “BECs”: explain this acronym here, that is the first time you use it in the manuscript body.

Line 279: I think you can eliminate the first “that”.

Line 349, “pvACef cells”: explain this acronym here, that is the first time you use it in the manuscript body (no matter if you have explained it in the footnotes of figure 5).

Line 371, small-vessel disease: use SVD, the acronym introduced in line 229-30.

Line 499: why don’t you use pvACef ?

Lines 576-7: pvACef has been already explained previously.

Line 601 and 604, “Pcfps … MMP”: explain these acronyms here, that is the first time you use them in the manuscript body (no matter if you have explained it in the footnotes of figure 16).

Line 635: cerebral blood flow = CBF.

Line 639: psACef already explained in line 582.

Lines 689-90: acronyms already explained previously.

Line 776, “Cerebral microbleeds (MBs)” = “Cerebral microbleeds-hemorrhages (CMBs)” in line 280. Decide one, and be consistent in using it.

There are a lot of acronyms, and the manuscript body is long (26 pages): think about to use a glossary to report the meanings of all the acronyms in alphabetical order. I think that it would help the reader.

Author Response

Response to Reviewer Number 3 Round 1 for MS ID: biomedicines-2791607

First the author would like to thank reviewer number 3 for the time, effort, and knowledge required to review this submitted manuscript.  Each of the suggested comments and recommended changes were helpful and will make this a better manuscript.  Please note that my changes to the submitted manuscript text (round 1) will be in blue lettering.

Comments and Suggestions for Authors

Comments and Suggestions for Authors

Impressive review: following are my few suggestion to improve the manuscript.

Author wishes to thank reviewer number 3 for this kind comment and the suggestions that were provided.  Each of these will help our readers to better understand the submitted manuscript.

Figure 1, footnotes: why do you use EC for brain endothelial cell, and later in the text you use BEC? Be consistent with the acronyms.

Authors response: ECs are now used consistently as follows: 

… In the true capillary, note how the perivascular astrocyte endfeet (pvACef) tightly abut and are directly adherent to the NVU mural cells brain endothelial cells (ECs), and pericytes foot processes (PcP - Pcfp) basement membrane(s) (BMs) via the pvACef dystroglycans….    And   … dAQP4 = dysfunction aquaporin 4 red lettering; EC = brain endothelial cell; EPVS = enlarged perivascular spaces; …

Line 122, “BBB”: explain this acronyms here, that is the first time you use it in the manuscript body (no matter if you have explained it in the footnotes of figure 4).

Authors response:  This has now been called out in the text as follows: … true capillary NVU with its blood-brain barrier (BBB) contains the normal PVS…

Line 183, “MetS and T2DM”: explain these acronyms here (not in line 225), that is the first time you use them in the manuscript body.

Authors response: … and clinical neurological disease states as they relate to obesity, metabolic syndrome (MetS) and type 2 diabetes mellitus (T2DM) (Tables 1, 2) [13, 14]…

Line 220, “BECs”: explain this acronym here, that is the first time you use it in the manuscript body.

Authors response: As follows: … In summary, the NVU consists of the following cellular components, which are neurons, perivascular astrocytes, microglia, pericytes, blood endothelial cell(s) (BECs), …

Line 279: I think you can eliminate the first “that”.

Authors response: The following changes are as follows:

     While we were unable to unravel the possible causes for these cerebral microbleeds ultrastructurally, we were able to demonstrate that the regions associated with microbleeds were definitely also associated with the detachment and separation of pvACef and there were no ultrastructural evidence of them being associated with congenital vascular malformations.

Line 349, “pvACef cells”: explain this acronym here, that is the first time you use it in the manuscript body (no matter if you have explained it in the footnotes of figure 5).

Authors response:  The following changes have been made in the submitted manuscript:  … The NVU and the NVU coupling are now well accepted and have received great interest in the field of neurobiology.  Its cells are comprised of BECs, Pcs, perivascular astrocyte endfeet (pvACef cells), …

Line 371, small-vessel disease: use SVD, the acronym introduced in line 229-30.

Authors response: The following changes have been made in the submitted manuscript, as follows;

… It is currently well recognized that most cases of LOAD have mixed vascular pathology and SVD [54, 55].  …

Line 499: why don’t you use pvACef ?

Authors response:  please see submitted manuscript as follows: … pvACef line the bulk of the brain vasculature at their abluminal surface [9]. ….

Lines 576-7: pvACef has been already explained previously.

Authors response:  the following changes have been made to the submitted manuscript as follows:  … which include (1) pvACef and occur primarily in cortical grey matter, (2) fibrous ACs,

Line 601 and 604, “Pcfps … MMP”: explain these acronyms here, that is the first time you use them in the manuscript body (no matter if you have explained it in the footnotes of figure 16).

Authors response:

… and separation from the NVU BEC and pericyte endfeet (Pcef) BMs and separate, ….  and

… oxidative stress (ROS) with matrix metalloproteinase(s) (MMPs)-2, 9 activation …

Line 635: cerebral blood flow = CBF.

Authors response:  the following changes have been made to the submitted manuscript, as follows:

… In summary, the NVU-PVU pvACef are responsible for neurovascular coupling, ensuring that regional CBF is tightly coupled to the activity of neurons …

Line 639: psACef already explained in line 582.

Authors response: the following changes have been made to the submitted manuscript, as follows:

… In summary, the NVU-PVU pvACef are responsible for neurovascular coupling, ensuring that regional CBF is tightly coupled to the activity of neurons in specific brain regions (Fig. 17A, B).  This dynamic regulation is essential for maintaining optimal brain function and responding to the varying metabolic demands of different brain areas. …

Lines 689-90: acronyms already explained previously.

Authors response: the following changes have been made to the submitted manuscript, as follows:

… In summary, the detachment-retraction of pvACef and psACef can both have widespread effects on the NVU and PSU respectively and potentially lead to disruptions in BBB integrity, …

Line 776, “Cerebral microbleeds (MBs)” = “Cerebral microbleeds-hemorrhages (CMBs)” in line 280. Decide one, and be consistent in using it.

Authors response: the following changes have been made to the submitted manuscript, as follows:

… CMBs are a common finding in patients with VSD, CAA, and LOAD [103, 120]. …

There are a lot of acronyms, and the manuscript body is long (26 pages): think about to use a glossary to report the meanings of all the acronyms in alphabetical order. I think that it would help the reader.

Author response:  I agree with reviewer number 3 and there already exists a list of abbreviations in the submitted manuscript as follows: 

Here is the list of abbreviations that is present in the submitted manuscript as follows:

Abbreviations: AC: astrocyte: ACef, astrocyte endfeet; AQP4, aquaporin-4; ATIII, BBB, blood–brain barrier; BEC(s), brain endothelial cell(s); BECact/dys, brain endothelial cell activation/dysfunction; BG, basal ganglia; BM(s), basement membranes; CAA, cerebral amyloid angiopathy; CBF, cerebral blood flow; CID, cognitive impairment and dysfunction; CL, capillary lumen; CMB(s), cerebral microbleed(s); CSF, cerebrospinal fluid; CSO, central semiovale; EPVS, enlarged perivascular spaces; DG. dystroglycan; EC, brain endothelial cell; EPVS, enlarged perivascular spaces; GS, glymphatic space; IPAD,  intramural periarterial drainage; ISF, interstitial fluid; ISS, interstitial space; LAN, lanthanum nitrate; LOAD, late-onset Alzheimer’s disease; LPS, lipopolysaccharide; MetS, metabolic syndrome; MGCs, microglia cells; MMP-2,-9, matrix metalloproteinase-2,-9; MRI, magnetic resonance imaging; MS, multiple sclerosis; PD, Parkinson’s disease; NVU, neurovascular unit-neuro-glia-vascular unit; Pc, pericyte; Pcfp, pericyte foot process; perivascular astrocyte endfeet; pvACef; perivascular astrocyte endfeet; psACendfeet, perisynaptic astrocyte endfeet; PVS, perivascular spaces; PVS/EPVS, perivascular space/enlarged perivascular space; rPVMΦ, resident perivascular macrophages; SAS, subarachnoid space; rPVMΦ, reactive perivascular macrophage; SVD, cerebral small vessel disease; T2DM, type 2 diabetes mellitus; TEM, transmission electron microscopy; TI/AJs, tight and adherens junctions; VAD, vascular dementia; VAT, visceral adipose tissue; VCID, vascular contributions to cognitive impairment and dementia; WMH, white matter hyperintensities.

Thank you very much!

Melvin R Hayden

Submitting author

Author response:  I agree with reviewer number 3 and there already exists a list of abbreviations in the submitted manuscript as follows: 

Here is the list of abbreviations that is present in the submitted manuscript as follows:

Abbreviations: AC: astrocyte: ACef, astrocyte endfeet; AQP4, aquaporin-4; ATIII, BBB, blood–brain barrier; BEC(s), brain endothelial cell(s); BECact/dys, brain endothelial cell activation/dysfunction; BG, basal ganglia; BM(s), basement membranes; CAA, cerebral amyloid angiopathy; CBF, cerebral blood flow; CID, cognitive impairment and dysfunction; CL, capillary lumen; CMB(s), cerebral microbleed(s); CSF, cerebrospinal fluid; CSO, central semiovale; EPVS, enlarged perivascular spaces; DG. dystroglycan; EC, brain endothelial cell; EPVS, enlarged perivascular spaces; GS, glymphatic space; IPAD,  intramural periarterial drainage; ISF, interstitial fluid; ISS, interstitial space; LAN, lanthanum nitrate; LOAD, late-onset Alzheimer’s disease; LPS, lipopolysaccharide; MetS, metabolic syndrome; MGCs, microglia cells; MMP-2,-9, matrix metalloproteinase-2,-9; MRI, magnetic resonance imaging; MS, multiple sclerosis; PD, Parkinson’s disease; NVU, neurovascular unit-neuro-glia-vascular unit; Pc, pericyte; Pcfp, pericyte foot process; perivascular astrocyte endfeet; pvACef; perivascular astrocyte endfeet; psACendfeet, perisynaptic astrocyte endfeet; PVS, perivascular spaces; PVS/EPVS, perivascular space/enlarged perivascular space; rPVMΦ, resident perivascular macrophages; SAS, subarachnoid space; rPVMΦ, reactive perivascular macrophage; SVD, cerebral small vessel disease; T2DM, type 2 diabetes mellitus; TEM, transmission electron microscopy; TI/AJs, tight and adherens junctions; VAD, vascular dementia; VAT, visceral adipose tissue; VCID, vascular contributions to cognitive impairment and dementia; WMH, white matter hyperintensities.

Thank you very much!

Melvin R Hayden

Submitting author

Reviewer 3 Report

Comments and Suggestions for Authors

A very interesting but very readable article about the recently described perivascular unit (PVU), which is located immediately adjacent to the true capillary neurovascular unit (NVU) in the postcapillary venule and contains normally benign perivascular spaces (PVS) and pathologically enlarged perivascular spaces (EPVS).

These structures are important because they were discovered as anatomical and functional structures, responsible for the removal of nervous tissue metabolites into the systemic circulation and form the glymphatic system.

The main goal of the analyzed article was to discuss the functions and structures of these systems and present their role in the development of neurological diseases (neurodegenerative, cerebrovascular disease, neuroinflammation) and their relationship with obesity, metabolic syndrome and type 2 diabetes.

In order to explain complex interactions, the article presents numerous, clear, although requiring a lot of attention, figures - although not everything is described in detail, e.g. 5 - abbreviation dACef - no explanation.

The authors presented numerous, comprehensive literature.

Author Response

Response to Reviewer Number 2 Round 1 for MS ID: biomedicines-2791607

First the author would like to thank reviewer number 2 for the time, effort, and knowledge required to review this submitted manuscript.  Each of the suggested comments and recommended changes were helpful and will make this a better manuscript.  Please note that my changes to the submitted manuscript text (round 1) will be in blue lettering.

Comments and Suggestions for Authors

A very interesting but very readable article about the recently described perivascular unit (PVU), which is located immediately adjacent to the true capillary neurovascular unit (NVU) in the postcapillary venule and contains normally benign perivascular spaces (PVS) and pathologically enlarged perivascular spaces (EPVS).

These structures are important because they were discovered as anatomical and functional structures, responsible for the removal of nervous tissue metabolites into the systemic circulation and form the glymphatic system.

The main goal of the analyzed article was to discuss the functions and structures of these systems and present their role in the development of neurological diseases (neurodegenerative, cerebrovascular disease, neuroinflammation) and their relationship with obesity, metabolic syndrome and type 2 diabetes.

In order to explain complex interactions, the article presents numerous, clear, although requiring a lot of attention, figures - although not everything is described in detail, e.g. 5 - abbreviation dACef - no explanation.

The authors presented numerous, comprehensive literature.

Authors response to reviewer number 2 comments and suggestions as follows;

Author has now placed in the legend to Figure 5 the following under abbreviations

dACef = dysfunctional astrocyte endfeet

Thank you for noting this deletion on my part.

Sincerely with gratitude,

Melvin R. Hayden

Round 2

Reviewer 1 Report

Comments and Suggestions for Authors

the authors have addressed my concerns.

Author Response

Thank you.